# Conserved enhancers control notochord expression of vertebrate *Brachyury*

Cassie L. Kemmler[1,21], Jana Smolikova[2,21], Hannah R. Moran[1,21], Brandon J. Mannion [3,4], Dunja Knapp [5], Fabian Lim [6,7,8], Anna Czarkwiani [5], Viviana Hermosilla Aguayo [9,10,11,12], Vincent Rapp[13], Olivia E. Fitch[14], Seraina Bötschi[15], Licia Selleri [9,10,11,12], Emma Farley [6,7], Ingo Braasch [14], Maximina Yun [5,16,17], Axel Visel [3,18,19], Marco Osterwalder [3,13,20], Christian Mosimann [1], Zbynek Kozmik[2] ✉ & Alexa Burger [1] ✉

The cell type-specific expression of key transcription factors is central to development and disease. *Brachyury/T/TBXT* is a major transcription factor for gastrulation, tailbud patterning, and notochord formation; however, how its expression is controlled in the mammalian notochord has remained elusive. Here, we identify the complement of notochord-specific enhancers in the mammalian *Brachyury/T/TBXT* gene. Using transgenic assays in zebrafish, axolotl, and mouse, we discover three conserved *Brachyury*-controlling notochord enhancers, *T3, C,* and *I*, in human, mouse, and marsupial genomes. Acting as Brachyury-responsive, auto-regulatory shadow enhancers, *in cis* deletion of all three enhancers in mouse abolishes Brachyury/T/Tbxt expression selectively in the notochord, causing specific trunk and neural tube defects without gastrulation or tailbud defects. The three *Brachyury*-driving notochord enhancers are conserved beyond mammals in the *brachyury/tbxtb* loci of fishes, dating their origin to the last common ancestor of jawed vertebrates. Our data define the vertebrate enhancers for *Brachyury/T/TBXTB* notochord expression through an auto-regulatory mechanism that conveys robustness and adaptability as ancient basis for axis development.

The defining feature of the chordate body plan is the notochord, a principal structure formed by the axial or chorda mesoderm that provides stability and rigidity along the body axis[1,2]. As mammals form an ossified spine, their notochord progressively regresses and its remnants form the nucleus pulposus within the intervertebral discs[3–7]. Notochord precursors emerge from the initial organizer and form in a stereotypical rostral-to-caudal trajectory as gastrulation proceeds, manifesting among the earliest visible structures in chordate embryos[1,8]. The deeply conserved T-box transcription factor gene *Brachyury* (also called *T* or *TBXT*) is a key regulator of notochord formation. Originally identified as dominant mutation *T* that caused short tails in mice, *Brachyury* expression and function has been linked to notochord emergence across chordates[9–15]. In addition to its central

role in notochord fate specification, the function of vertebrate *Brachyury* is required for proper primitive streak formation, tailbud specification, and subsequent neuromesodermal progenitor control[16–18]. However, how the expression of this central developmental transcription factor is selectively regulated to achieve its notochord activity in mammals remains unresolved.

The central contribution of the notochord and the tailbud to different morphological adaptions and locomotion strategies shows in the diversification of axial structures across vertebrates[19]. Gain and loss of gene copies and of their associated gene-regulatory elements are major drivers of evolutionary innovation, and the *Brachyury* gene family itself is a prime example of this process. *Brachyury* predates the origin of, and was present as, a single copy gene in the chordate

ancestor[20,21]. Following two whole genome duplications in early vertebrates and the subsequent loss of one of four *Brachyury* paralogs, three gene paralogs were present in the jawed vertebrate ancestor: *Tbxta*, *Tbxtb*, and *Tbx19*[21]. *Tbxta* became subsequently lost within the tetrapod lineage, resulting in mammals and birds ultimately only retaining *Tbxtb* (commonly called *Brachyury/T/TBXT* in tetrapods including humans)[22]. In contrast, ray-finned fishes retained both *tbxta/ntla* and *tbxtb/ntlb*, the latter being the ortholog of the remaining human *Brachyury/T/TBXT* (*de facto TBXTB*) gene[17].

Curiously, *tbxta/ntla* has become the predominant functional *Brachyury/T/TBXT* gene in zebrafish, as documented in classic mutants for *ntla* (*no tail a*) that fail to form a tail and notochord[13,15]. While no mutant for zebrafish *tbxtb/ntlb* has been reported to date, morpholino-based knockdown studies indicate that *tbxtb* function adds minimally to the dominant role of zebrafish *tbxta*[17]. This variable copy number of *Brachyury* genes across vertebrates came along with selection and divergence of regulatory elements controlling *Brachyury* gene expression during distinct developmental timepoints and cell types. Promoter-proximal regions in the Ciona *Brachyury* gene and in the zebrafish *tbxta* gene drive early organizer and notochord activity[10,23]. In contrast, the promoter-proximal region called *Tstreak* of *Brachyury/T/Tbxtb* in mouse, human, and *Xenopus* has previously been found to drive primitive streak expression in response to canonical Wnt/beta-catenin signaling, yet lacks any notochord-driving activity[24-26]. Further, recent work documented that deleting a large 37 kb-spanning region upstream of mouse *Brachyury/T/Tbxtb* leads to mutant phenotypes consistent with a selective loss of *Brachyury* notochord expression[27]. A small element termed *TNE* in the 37 kb interval was sufficient to drive specific notochord expression in mouse reporter assays, yet its deletion showed mild to no phenotypic consequences[27]. These pioneering data show that additional regulatory element(s) in addition to *Tstreak* and *TNE* contribute to *Brachyury/Tbxtb* expression specifically in the notochord. Uncovering the regulation of the vertebrate *Brachyury* notochord enhancer(s) will expand our understanding of the evolutionary history of this key developmental regulator and of the mechanisms leading to notochord formation. In particular, comparison to the Ciona *Brachyury* locus containing two upstream shadow enhancers with well-defined regulatory grammar[28,29] may inform *cis*-regulatory adaptations at the onset of vertebrate emergence.

Uncovering the regulatory elements responsible for its notochord expression also promises to shed light onto the role of *Brachyury* in adult human spine health and in chordoma tumors, a rare sarcoma of the spine that is hypothesized to arise from notochord remnants[30-32]. Several familial chordomas harbor duplications or further complex amplifications of the *Brachyury/T/TBXTB* locus that possibly convey chordoma susceptibility to carriers[33-35]. These findings suggest that chordoma-associated *Brachyury/T/TBXTB* locus amplifications contain, or hijack the action of, *cis*-regulatory elements to possibly drive *Brachyury/T/TBXTB* expression in chordoma, potentially with *Brachyury* controlling its own expression as indicated by ChIP-seq findings[36].

Here, we identify the complement of three auto-regulated shadow enhancers *T3*, *C*, and *I* in the *Brachyury/T/Tbxtb* locus that convey notochord activity. We combined (i) genomic data from human chordoma tumor cell lines, human embryonic stem cells, and mouse embryonic stem cells; (ii) non-coding element conservation across mammals (human, mouse, *Monodelphis*) and all vertebrates; (iii) transgenic reporter assays in zebrafish, mouse, axolotl, and Ciona; (iv) and enhancer knockouts in mice. In triple enhancer knockout mice, we document the selective absence of Brachyury protein in the notochord and subsequent neural tube and trunk defects as linked to notochord perturbations. Using comparative genomics, we uncover that the location and activity of the enhancers *T3*, *C*, and *I* is conserved within the *Brachyury/tbxtb* loci across jawed vertebrates. Our data uncover a deep conservation of shadow enhancers regulating *Brachyury*

expression in the notochord, one of the most prominent developmental structures of the vertebrate body and involved in spine and neural tube defects.

## Results

### Defining a region for human *Brachyury* notochord expression

To identify enhancer elements with notochord activity in the human *Brachyury/T/TBXTB* locus, we analyzed the *Brachyury/T/TBXTB* locus to narrow down a minimally required genomic region around the *Brachyury* gene body. Familial and sporadic chordoma feature duplications and/or complex amplifications of *Brachyury*[33-35,37], suggesting that essential *cis*-regulatory elements for notochord expression lie within the commonly amplified region. Available genomic patient data outlined a minimally amplified region of ~50 kb surrounding the human *Brachyury* gene body, with individual tumors extending their amplifications proximal or distal of this minimal region[34,37] (Fig. 1A). Within this minimal interval and its vicinity, we uncovered several regions that have been charted as open chromatin in the chordoma cell lines U-CH2 and MUGCHOR using ATAC-seq[36,38], indicating potential regulatory elements in accessible chromatin, including a super-enhancer region previously proposed to be active in chordoma[38] (Fig. 1A). Further, mammalian *Brachyury* has been postulated to control its own notochord expression[27,39]. Using *Brachyury/T* ChIP-seq data from the human chordoma tumor cell line U-CH1 and human ES-derived mesoderm progenitor cells[36,40], we found discrete Brachyury binding events within the minimal amplification interval and its vicinity (Fig. 1A). Genome alignment of human versus other mammalian species indicated candidate enhancer regions (conserved non-coding elements; CNEs) through non-coding sequence conservation in mouse and the more distant marsupial *Monodelphis domestica*[41] (Fig. 1A).

From our combined locus analysis, we identified the six initial candidates *T3*, *K*, *J*, *C*, *I*, and *L* as putative notochord enhancer elements in the vicinity of the human *Brachyury* gene (Fig. 1A, Supplementary Data 1; all Supplementary Data is included in the Supplementary Information file). While *K* and *J* represent conserved sequence to other mammalian genomes, candidates *I* and *L* notably lie in the annotated chordoma super-enhancer region[38]. From this combined analysis, we hypothesized that individual or combined elements among the six enhancer candidates could convey notochord activity to the human *Brachyury* gene.

### *Brachyury* enhancers have autonomous notochord activity

Given the evolutionarily conserved notochord expression of vertebrate Brachyury genes, we hypothesized that the human enhancers may be correctly interpreted in a model vertebrate. We initially tested all six enhancer candidates in zebrafish that allows for highly efficient reporter gene activity screening in developing embryos. To test their activity within a broad evolutionary framework, we cloned the human enhancer element candidates *T3*, *K*, *J*, *C*, *I*, and *L* into reporter vectors coupled with the mouse *betaE-globin* minimal promoter to express the blue fluorophore *mCerulean* for enhancer testing in zebrafish embryos[42]. Upon co-injection into one cell-stage zebrafish embryos together with *ubi:mCherry* as injection control, the human *hs_T3*, *hs_C*, and *hs_I* elements resulted in *mCerulean* expression in the developing zebrafish notochord during early somitogenesis, followed by strong, selective notochord activity in injected embryos at 24 h post-fertilization (hpf) ($n = 32/61$, $n = 155/227$, $n = 76/117$; mCerulean-positive notochord/total mCherry-positive embryos) (Fig. 1B–D, Supplementary Data 2). Zebrafish embryos injected with *hs_T3*, *hs_C*, and *hs_I* reporters maintained notochord-specific *mCerulean* expression throughout our observations until 5 days post-fertilization (dpf). In contrast, we did not observe any specific *mCerulean* reporter expression at any timepoint with elements *hs_K*, *hs_J*, and *hs_L* ($n = 0/68$, $n = 0/63$, $n = 0/254$) (Supplementary Data 2). Notably, *hs_C* was still active when further trimming the sequence 5′ and 3′

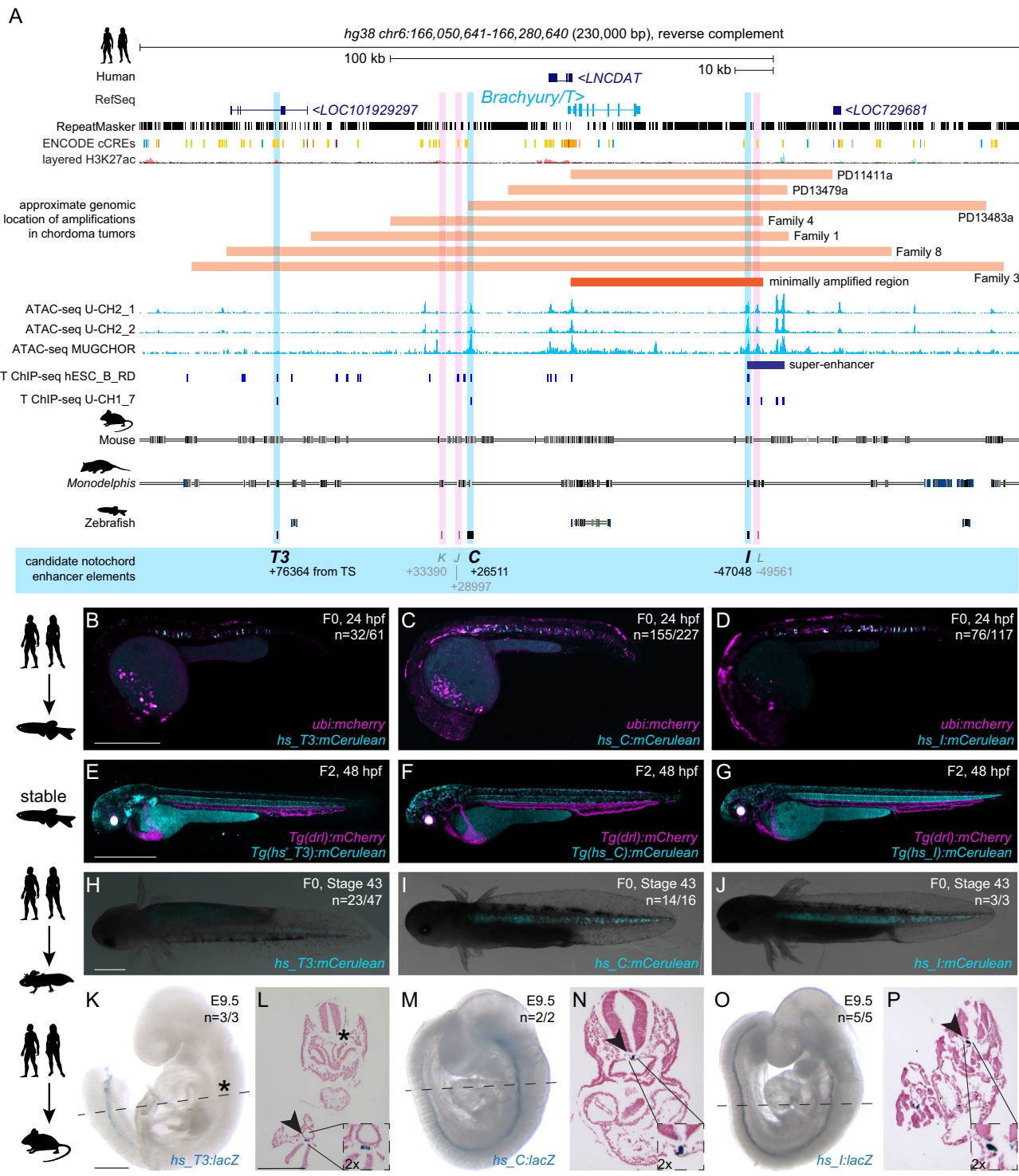

(*hs_Cshort*, *n* = 55/103) (Supplementary Fig. 1A–C, Supplementary Data 2). Germline-transmitted, stable transgenic integrations for *mCerulean* reporters based on *hs_T3*, *hs_C*, and *hs_I* recapitulated the transient reporter results and consistently showed selective notochord expression, with minimal variability across independent transgenic insertions for each enhancer reporter (followed to at least F3 generation) (Fig. 1E–G). Together, these data indicate that the three enhancer elements *hs_T3*, *hs_C*, and *hs_I* within the human *Brachyury/T/TBXTB* locus convey notochord activity when tested in zebrafish.

Next, we tested the activity of *hs_T3*, *hs_C*, and *hs_I* in axolotl (*Ambystoma mexicanum*) as a representative amphibian species[43,44]. Upon microinjection, reporters based on *hs_T3*, *hs_C*, and *hs_I*

enhancer elements showed consistent reporter expression in the notochord of axolotl embryos (*n* = 23/47, *n* = 14/16, *n* = 3/3) throughout tailbud stages (st. 30-41) and beyond (Fig. 1H–J, Supplementary Fig. 1D–M, Supplementary Data 2). Notably, 50% of *hs_T3*-positive F0 animals had additional expression in other mesodermal tissues such as trunk muscles. In contrast, 80% and 100% of positive *hs_C* and *hs_I* F0 animals, respectively, showed expression exclusively in the notochord. In addition, *hs_C* and *hs_I* reporter expression was distributed along the entire rostral-caudal axis in all observed embryos, while *hs_T3* reporter expression was frequently restricted to more caudal portions of the notochord. Combined, these results indicate that the human enhancers *hs_T3*,

**Fig. 1 | Human *Brachyury* enhancer elements *T3, C,* and *I* are active in different species. A** Human *Brachyury/T/TBXTB* locus with surrounding gene loci adapted from UCSC genome browser. Repeats marked in black using the RepeatMasker track; additional tracks include the ENCODE conserved *cis*-regulatory elements (cCREs) and layered H3K27ac signals. Further annotated are approximate amplifications (light orange) and the minimally amplified region (dark orange) in chordoma tumors. ATAC-sequencing (light blue peaks) and T ChIP-sequencing (dark blue lines) suggest enhancer elements (light pink highlight, not active; light blue highlight, active) that are conserved in mouse and the marsupial *Monodelphis domestica*. **B–D** Representative F0 zebrafish embryos injected with the human enhancer elements *hs_T3* (**B**), *hs_C* (**C**), and *hs_I* (**D**) showing mosaic *mCerulean* reporter expression in the notochord at 24 hpf and expression of *ubi:mCherry* as injection control. N represents the number of animals expressing mCerulean in the notochord relative to the total number of animals expressing mCherry. Scale bar in (**B**): 0.5 mm, applies to **B**, **C**. **E–G** Representative images of stable transgenic F2 embryos at 48 hpf for each of the human enhancer elements *hs_T3*, *hs_C*, and *hs_I* crossed to *Tg(drl:mCherry)* that labels lateral plate mesoderm and later cardiovascular lineages. Transgenic F2 embryos recapitulate the F0 expression pattern in the notochord, with *hs_T3* (**E**) additionally expressing mCerulean in the pharyngeal arches and fin, and *hs_I* (**G**) in the proximal kidney close to the anal pore. Enhancer

element *hs_C* (**F**) stable transgenic lines have lower relative notochord reporter activity than *hs_T3* and *hs_I*. Scale bar in (**E**): 0.5 mm, applies to **E**–**G**. **H–J** Representative F0 axolotl embryos at peri-hatching stages expressing mCerulean from the human enhancers *hs_T3* (**G**), *hs_C* (**H**), *hs_I* (**I**). N represent the number of animals expressing mCerulean in the notochord relative to the total number of animals showing any mCerulean expression. Scale bar in (**H**): 1 mm, applies to **H**–**J**. **K, M,** and **O** Representative images of transgenic E9.5 mouse embryos expressing *lacZ* (encoding beta-galactosidase) under the human enhancers *hs_T3* (**K**), *hs_C* (**M**), and *hs_I* (**O**) visualized with X-gal whole-mount staining. While *hs_C* and *hs_I* express beta-galactosidase in the entire notochord, beta-galactosidase expression from *hs_T3* is restricted to the posterior notochord. Black asterisk marks absence of beta-galactosidase in the anterior notochord. N represent the number of animals expressing beta-galactosidase in the notochord relative to the total number of animals with tandem integrations at *H11*. Dotted lines represent the sectioning plane. Scale bar in (**K**): 0.5 mm, applies to (**K, M, O**). **L, N, P** Representative images of Fast Red-stained cross sections from embryos shown on the left, *hs_T3* (**L**), *hs_C* (**N**), and *hs_I* (**P**). Black arrowheads point at notochord, and inserts show notochords at 2x higher magnification. Scale bar in (**L**): 0.25 mm, applies to **L, N, P**. The species silhouettes were adapted from the PhyloPic database (www.phylopic.org).

*hs_C,* and *hs_I* also integrate regulatory input for driving notochord activity in amphibians.

We next tested if human enhancers *hs_T3, hs_C,* and *hs_I* also drive notochord-specific reporter activity in mouse embryos. For increased specificity and reproducibility, we used a site-directed transgenic integration strategy at the *H11* locus (enSERT)[45] to generate mouse embryos harboring *enhancer-LacZ* reporter transgenes. As observed in zebrafish and axolotl, *hs_T3, hs_C,* and *hs_I* elements exhibited specific and selective notochord expression in mouse embryos at E9.5 ($n = 3/3$, $n = 2/2$ and $n = 5/5$) (Fig. 1K, M, O, Supplementary Data 2). Of note, *hs_T3* reporter activity appeared predominantly confined to the posterior notochord compared to *hs_C* or *hs_I*, which showed reporter activity in the entire mouse notochord. Histological analysis of Nuclear Fast Red-stained transversal sections from transgenic mouse embryos further confirmed reproducible, notochord-specific activity for human notochord enhancer elements *hs_T3, hs_C,* and *hs_I* (Fig. 1L, N, P).

Taken together, we identified three enhancer candidates in the human *Brachyury/T/TBXTB* locus, that all display notochord enhancer activity as transgenic reporters when tested in teleost fish, amphibian, and rodent embryos, suggesting pan-bony vertebrate activity and function.

**Dependence of human *Brachyury* enhancers on T-box motifs**
Published ChIP-seq data indicated Brachyury binding at *hs_T3, hs_C,* and *hs_I* (Fig. 1A), suggesting that notochord expression of the *Brachyury/T/Tbxtb* gene might be auto-regulated by Brachyury itself[27,39]. We investigated if the three human notochord enhancer elements contained a TBXT binding motif (short T-box, Fig. 2A) using FIMO[46]. We found that enhancer element *hs_T3* contained two low *p*-value T-box motifs, enhancer element *hs_I* contained one low *p*-value T-box motif, whereas enhancer element *hs_C* contained two possibly degenerate T-box motifs that we only identified when significantly increasing the *p*-value (Fig. 2B), with two additional T-box motifs with even higher *p*-values that we did not further pursue in this work (Supplementary Fig. 2A, B). We then generated the reporter constructs *hs_T3ΔTbox:mApple, hs_CshortΔTbox:mApple,* and *hs_IΔTbox:mApple* in which we deleted the respective T-box motifs in the enhancer elements, as well as constructs containing the wildtype enhancer elements in an identical backbone (Fig. 2C). The reporter constructs further harbored the transgenesis marker *exorh:EGFP* (expression in the pineal gland, Fig. 2D–I) for precise quantification of reporter activity[42]. After injection into zebrafish embryos and in line with the enhancer element activity at 24 hpf (Fig. 1B–D), we observed continued and reproducible notochord expression at 48 hpf with all three wild-type enhancer element reporters *hs_T3:mApple, hs_C:mApple,* and

*hs_I:mApple* ($n = 42/58$, $n = 39/57$ and $n = 62/79$; mCerulean-positive notochord/total EGFP pineal gland-positive embryos) (Fig. 2D, F, H, Supplementary Data 2). However, we observed a complete loss of specific notochord reporter activity in zebrafish embryos injected with the deletion constructs *hs_T3ΔTbox:mApple, hs_CshortΔTbox:mApple,* and *hs_IΔTbox:mApple* ($n = 6/113$, $n = 7/53$, $n = 1/41$), with positive embryos containing few labeled notochord cells (Fig. 2E, G, I, Supplementary Data 2). In contrast, individual deletion of the high *p*-value T-box motifs in enhancer element *hs_C* did not result in significant loss of reporter activity ($n = 28/50$, $n = 15/63$, Supplementary Fig. 2C, D).

Together, we conclude that the T-box motifs in the notochord enhancers *hs_T3, hs_C,* and *hs_I* are critical to the activity of these regulatory elements in our reporter assays. These data support the model in which *Brachyury/T/TBXTB* auto-regulates its own expression in the notochord through a defined motif in its notochord-regulatory elements[27,39].

**Brachyury notochord enhancers are conserved across mammals**
We next sought to determine if other mammalian genomes harbor orthologous *T3, C,* and *I* enhancer regions in their *Brachyury/T/Tbxtb* loci. Here, we focused on the orthologous *T3, C,* and *I* enhancer candidate regions from mouse (Fig. 3A). As in the human *Brachyury/T/TBXT*B locus, we found open chromatin and Brachyury protein binding events at the mouse orthologs of the putative enhancer elements *T3, C,* and *I,* as well as the well-characterized murine *Brachyury/T/Tbxtb* promoter *Tstreak* (Fig. 3A).

When transiently tested in zebrafish, both mouse enhancer *mm_T3* and *mm_I* showed reporter activity emerging arbitrarily throughout the gastrulating embryo at around 6 hpf (50% epiboly, shield stage) (Supplementary Fig. 3A–D), before expression became restricted to the developing notochord ($n = 46/67$, $n = 61/66$) at 24 hpf (Fig. 3B, D, Supplementary Data 2). Of note, our mouse enhancer *mm_T3* contains the previously identified element *TNE*, which has been established to act as autonomous notochord enhancer when tested in mouse embryos and gastruloid cultures[27]. In contrast, mouse enhancer *mm_C* failed to drive any reporter expression in the zebrafish notochord ($n = 0/88$) (Fig. 3C, Supplementary Data 2). Imaging transgenic zebrafish carrying mouse *mm_I* as stable reporter documented robust notochord expression, again with little variability across independent transgenic insertions (Supplementary Fig. 3E). In contrast, the murine *Brachyury/T/Tbxtb* promoter region *Tstreak*[24–26] showed transient, variable reporter expression in the zebrafish shield at around 6 hpf, with no reporter activity upon somitogenesis ($n = 79/102$) (Supplementary Data 2). We further tested the mouse ortholog of enhancer candidate *mm_J*, as well as the two lesser conserved elements *mm_T1*

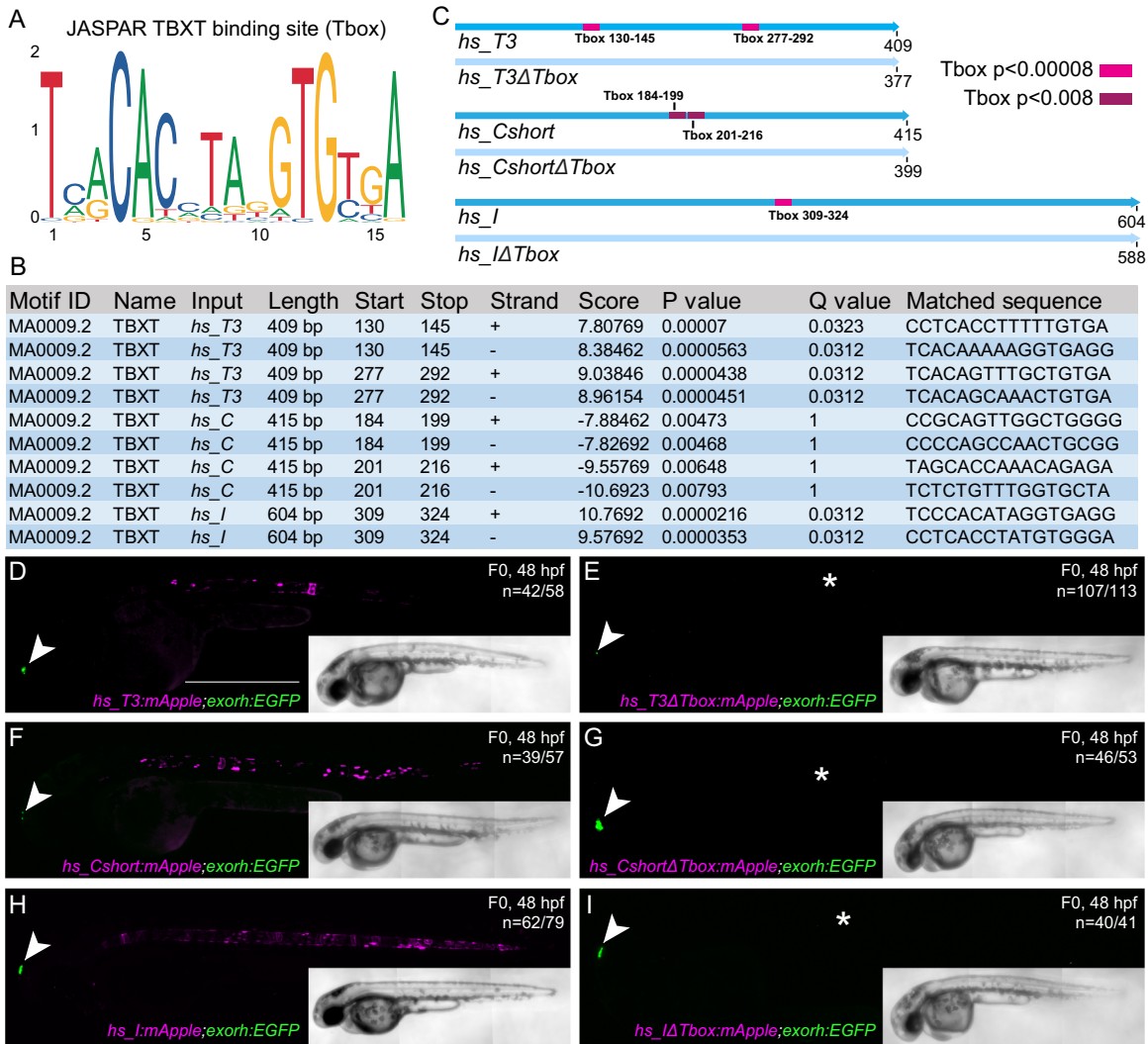

**Fig. 2 | Identified TBXT binding sites in the enhancer elements are essential for reporter activity. A** Sequence of the human TBXT binding site (T-box) using JASPAR. **B** FIMO output with location of the T-box, statistical significance, and matched sequence within the enhancer elements. *P*-values were calculated by FIMO which computes a log-likelihood ratio score for each position in the sequence, then converts this score to a *P*-value, and then applies false discovery rate analysis to estimate a *Q*-value for each position. **C** Schematic depiction of the wildtype human enhancer elements with the TBXT binding site/T-box (pink, red, purple boxes) and the enhancer elements without the respective T-box sites (*ΔTbox*). The human enhancer elements are depicted in the reverse complement direction. Tbox130-145, Tbox277-292, Tbox309-324: *p* < 0.00008, Tbox184-199: *p* < 0.005, Tbox201-216: *p* < 0.008. **D–I** Injection of the wildtype enhancer elements *hs_T3* (**D**), *hs_Cshort* (**F**), *and hs_I* (**H**) as reporter constructs results in mApple fluorophore expression in the notochord at 48 hpf, whereas injection of *hs_T3ΔTbox* (**E**), *hs_CshortΔTbox* (**G**), and *hs_IΔTbox* (**I**) show complete loss of notochord expression (asterisks in **E**, **G**, **I**). Arrowheads (**D–I**) mark EGFP expression in the pineal gland from the transgenesis marker *exorh:EGFP*. Scale bar in (**D**): 0.5 mm, applies to **D–I**.

and *mm_T5*, none of which showed reporter activity in zebrafish embryos up to 5 dpf (*n* = 0/98, *n* = 0/98, *n* = 0/79) (Supplementary Data 2).

Tested with site-directed reporter transgenesis at *H11*, *mm_T3* and *mm_I* conveyed specific notochord activity in mouse embryos at E9.5 (*n* = 2/2, *n* = 4/4) (Fig. 3E, G, Supplementary Data 2). In contrast, and consistent with our observations in zebrafish reporter assays, *mm_C* did not show any detectable reporter activity in the notochord in mouse embryos at E9.5 (*n* = 0/2) (Fig. 3F, Supplementary Data 2).

While humans and mice diverged ~90 million years ago, marsupials split from eutherians (placental mammals) ~160 million years ago[41,47–50]. The opossum *Monodelphis domestica* is a representative marsupial species and provides a more distant comparative species to human and mouse (Supplementary Fig. 4A). Detailed sequence alignments documented dispersed conserved regions along the entire sequences for all three enhancer candidates in Monodelphis (Fig. 4A). When injected into zebrafish embryos as *mCerulean* reporters, the Monodelphis-derived *md_T3*, *md_C*, and *md_I* enhancer element candidates all conveyed specific notochord activity at 24 hpf (*n* = 47/62, *n* = 142/184, *n* = 74/97) (Fig. 4B–D, Supplementary Data 2). Similar to the mouse elements, *md_T3* transiently started reporter expression at around 6 hpf (Supplementary Fig. 4B, C), whereas *md_C* and *md_I* started to be active at early somitogenesis, similar to the human ones. In addition to the notochord activity, *md_C* reporter-injected zebrafish embryos showed transient reporter expression in the heart whereas *md_I* reporter-injected embryos showed transient expression in the brain and spinal cord neurons (Fig. 4C, D).

Given the mammalian sequence conservation and differential responses in reporter assays, we next tested the notochord enhancer element candidates in the tunicate *Ciona intestinalis* as non-vertebrate outgroup. As a chordate, Ciona forms a bona fide notochord[51]. Testing *T3*, *C*, and *I* of human, mouse, and *Monodelphis* by reporter gene assays in Ciona, we found that only *Monodelphis*-derived *md_C* showed specific and robust reporter activity in the notochord (*n* = 119/150)

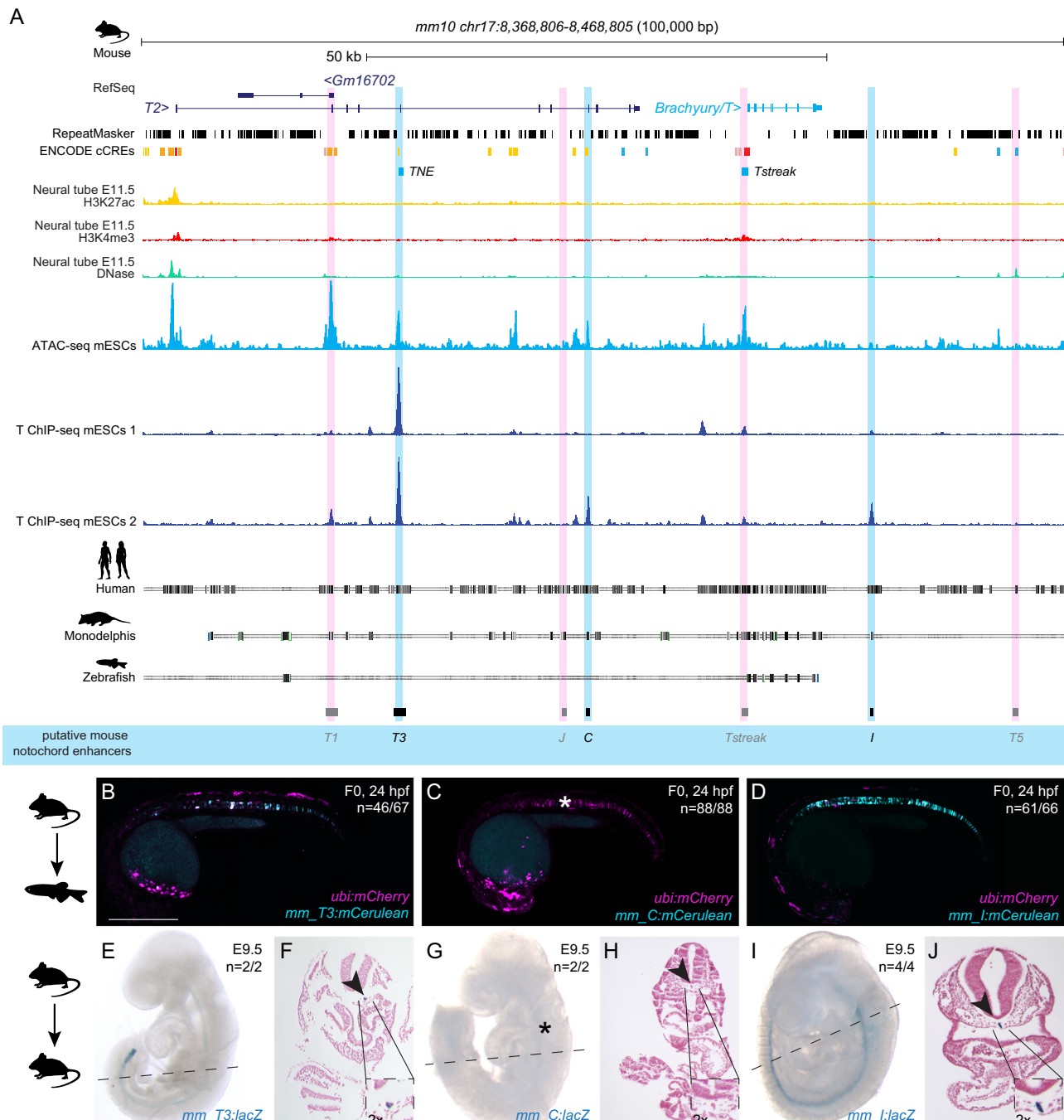

**Fig. 3 | Mouse *Brachyury* enhancer elements are active in different species.**
**A** Mouse *Brachyury/T/TBXTB* locus adapted from the UCSC genome browser. Repeats marked in black using the RepeatMasker track; additional tracks include the ENCODE cCREs, H3K27ac (yellow), H3K4me (red) and DNase (green) signals. ATAC-sequencing (light blue peaks) and T ChIP-sequencing (dark blue lines) indicate enhancer elements (light pink highlight, not active; light blue highlight, active) that are conserved in human and *Monodelphis*. **B–D** Representative F0 zebrafish embryos injected with the mouse enhancer elements *mm_T3* (**B**), *mm_C* (**C**), and *mm_I* (**D**). *mm_T3* and *mm_I* show mosaic *mCerulean* reporter expression in the notochord at 24 hpf and mosaic *ubi:mCherry* expression as injection control. Mouse enhancer element *mm_C* is not active in the zebrafish notochord (asterisk in **C**). N represent the number of animals expressing mCerulean in the notochord relative to the total number of animals expressing mCherry. Scale bar in (**B**): 0.5 mm, applies to

(**B–D**). **E**, **G**, **I** Representative images of transgenic E9.5 mouse embryos expressing *lacZ* (encoding beta-galactosidase) under the mouse enhancer elements *mm_T3* (**E**), *mm_C* (**G**) and *mm_I* (**I**) visualized with X-gal whole mount staining. While *mm_T3* and *mm_I* express beta-galactosidase in the entire notochord, beta-galactosidase expression from mouse *mm_C* is absent (asterisk in **G**). N represent the number of animals expressing beta-galactosidase in the notochord relative to the total number of animals with tandem integrations at *H11*. Dotted lines represent the sectioning plane. Scale bar in (**E**): 0.5 mm, applies to **E**, **G**, **I**. **F**, **H**, **J** Representative images of Fast Red-stained cross sections from embryos shown on the left, *mm_T3* (**F**), *mm_C* (**H**), and *mm_I* (**J**). Black arrowheads point at notochord, and inserts show notochords at 2x higher magnification. Scale bar in **F**: 0.25 mm, applies to **F**, **H**, **J**. The species silhouettes were adapted from the PhyloPic database (www.phylopic.org).

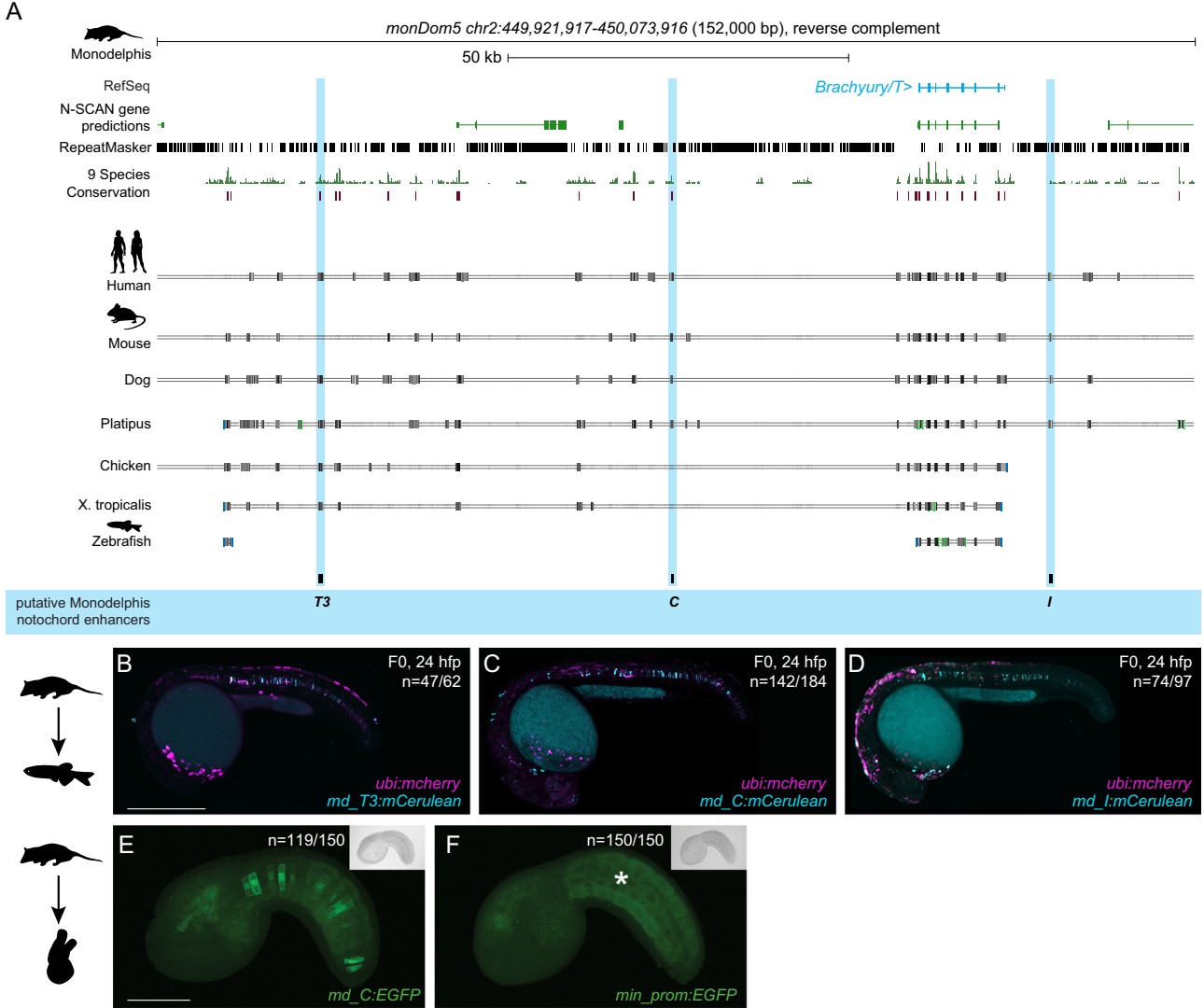

**Fig. 4 | Monodelphis *Brachyury* enhancer elements are active in different species. A** Monodelphis *Brachyury/T/TBXTB* locus adapted from the UCSC genome browser. Repeats are marked in black using the RepeatMasker track. Further annotated are tracks containing N-SCAN gene predictions and 9 Species Conservation. The light blue highlighted boxes mark the Monodelphis enhancer elements *T3*, *C* and *I* and their conservation in other species. **B–D** Representative F0 zebrafish embryos injected with the Monodelphis enhancer elements *md_T3* (**B**), *md_C* (**C**), and *md_I* (**D**) showing mosaic *mCerulean* reporter expression in the zebrafish notochord at 24 hpf. *ubi:mCherry* was used as injection control. N represent the number of animals expressing mCerulean in the notochord relative to the total number of animals expressing mCherry. Scale bar in (**B**): 0.5 mm, applies to (**B**, **C**). **E**, **F** Representative images of Ciona embryos electroporated with Monodelphis enhancer element *md_C* (**E**), and minimal *forkhead* promoter (*fkh*) only as control (**F**). Monodelphis enhancer element *md_C* expresses EGFP throughout the entire Ciona notochord, compared to minimal *fkh* promoter only which does not express EGFP at all (asterisk in **F**). *N* represent the number of animals expressing EGFP in the notochord relative to the total number of animals. Inserts on the top right represent bright field images of respective embryos. Scale bar in (**E**): 0.05 mm, applies to **E**, **F**. The species silhouettes were adapted from the PhyloPic database (www.phylopic.org).

compared to all other eight elements (*n* = 0/150) and minimal promoter only control (*n* = 0/150) (Fig. 4E, F, Supplementary Data 2).

Taken together, and extending previous work on the mouse *TNE* element[27], our data indicate that three distant elements in the mammalian *Brachyury/T/Tbxtb* locus with differential activity converge on providing notochord-specific activity in reporter assays across chordates.

### Enhancer deletions cause selective loss of *Brachyury* in mice

While especially enhancer element *C* seems to have diverged in activity (or is sensitive to the specific *trans* environment it was tested in), all three elements *T3*, *C*, and *I* remain conserved and detectable at the sequence level throughout the mammalian clade. In mice, homozygous *Brachyury/T/Tbxtb* mutations in the gene body cause post-implantation defects leading to embryonic lethality between E9.5 and

E10.5[52–54]. Previous work established that deletion of mouse enhancer *TNE* does not cause a fully penetrant loss of *Brachyury/T/Tbxtb* expression in the developing notochord, indicating the presence of additional shadow elements interacting with, or compensating for, *TNE*[27]. To functionally test if the three enhancer elements are involved in *Brachyury/T/Tbxtb* expression in the mouse notochord, we generated a series of knockout alleles targeting the three mouse enhancer elements *T3*, *C*, and *I* (Fig. 5A).

We employed CRISPR-Cas9 genome editing using target sites flanking the enhancers and established heterozygous and homozygous mice carrying individual and combined enhancer deletions (Fig. 5A, Supplementary Fig. 5A). Compared to E9.5 wildtype control embryos (Fig. 5B) (*n* = 14/14), neither homozygous deletion of mouse *C* ($T^{\Delta C/\Delta C}$) (*n* = 7/7) or *I* ($T^{\Delta I/\Delta I}$) (*n* = 7/7) alone, nor heterozygous ($T^{+/\Delta C,I}$) (*n* = 12/12), heterozygous ($T^{+/\Delta T3}$) (*n* = 7/7) (Supplementary Fig. 5B–F) or

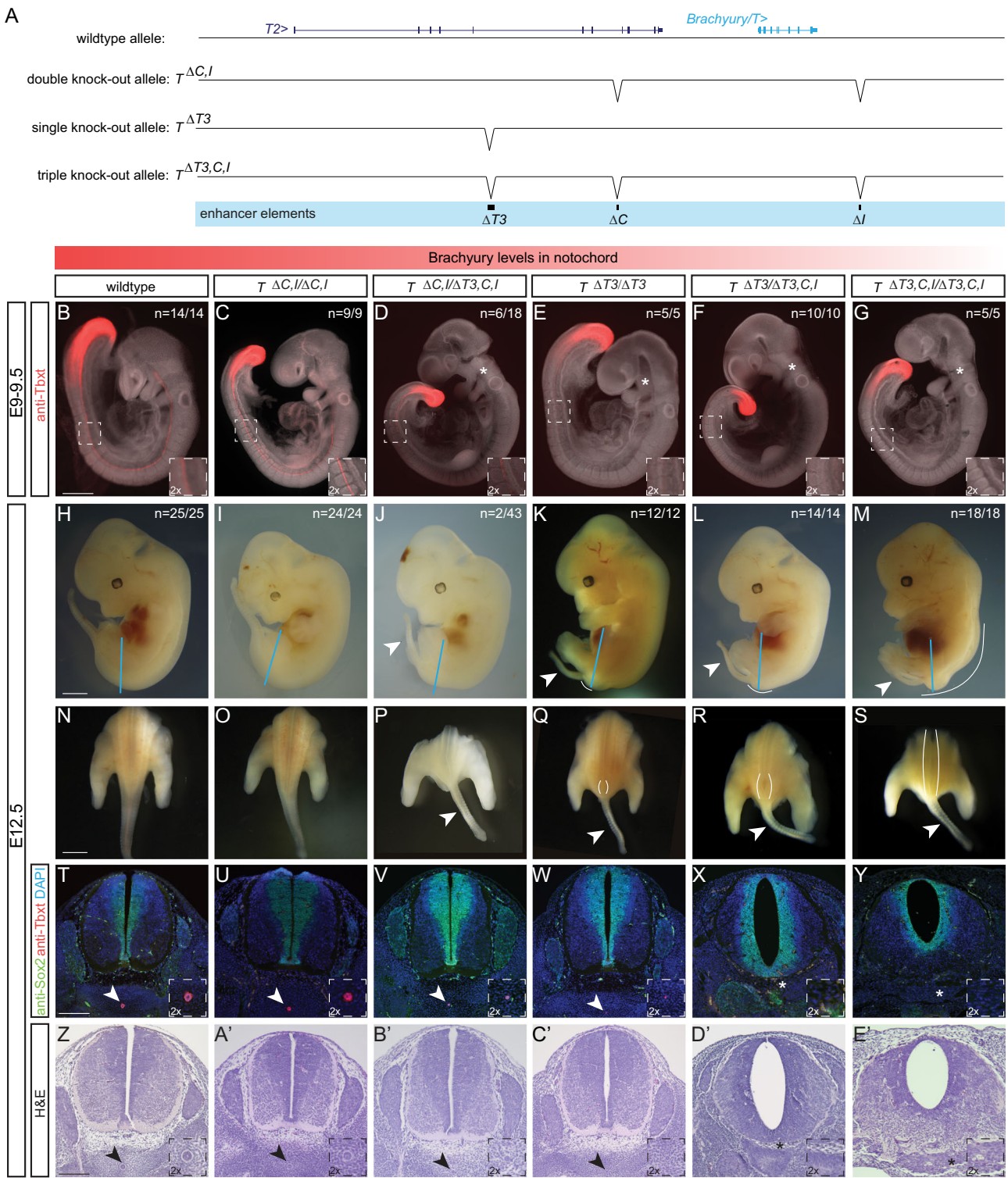

Nature Communications | (2023) 14:6594

homozygous deletion of both *C* and *I* ($T^{\Delta C,I/\Delta C,I}$) (n = 9/9) (Fig. 5C) altered *Brachyury/T/Tbxtb* expression in the notochord as determined by Brachyury/T antibody staining.

In contrast, we observed reduced Brachyury/T/Tbxtb expression in the notochord of E9.5 embryos in a dose-dependent manner when we combined *ΔT3* with *ΔC,I* deletions. E9.5 embryos heterozygous for the triple knockout chromosome carrying *ΔT3,C,I* ($T^{+/\Delta T3,C,I}$) *in cis* appeared normal (n = 14/14) (Supplementary Fig. 5F). In contrast, in trans-heterozygous E9.5 embryos carrying *ΔC,I* and *ΔT3,C,I* alleles ($T^{\Delta C,I/\Delta T3,C,I}$), we documented reduced Brachyury/T/Tbxtb protein in the

caudal portion of the notochord in all embryos (n = 18/18) with individual embryos also displaying reduced or lost Brachyury/T/Tbxtb protein in the trunk and rostral portion (n = 6/18) (Fig. 5D). Similarly, in E9.5 embryos homozygous for *ΔT3* ($T^{\Delta T3/\Delta T3}$) (n = 5/5) (Fig. 5E), we observed reduced Brachyury/T/Tbxtb protein levels, as previously reported for homozygous *TNE* embryos[27]. Brachyury/T/Tbxtb protein levels were even further reduced or lost in the entire notochord of trans-heterozygous for *ΔT3* and *ΔT3,C,I* alleles ($T^{\Delta T3/\Delta T3,C,I}$) (n = 10/10) (Fig. 5F). These data are consistent with, and expand upon, previous observations that the severity of *Brachyury/T/Tbxtb* phenotypes

**Fig. 5 | Deletion of the three enhancer elements *T3, C* and *I* results in selective loss of Brachyury protein expression in the notochord at E9.5 and posterior defects at E12.5. A** Overview of wildtype mouse *Brachyury/T/TBXTB* locus adapted from the UCSC genome browser and deletion alleles generated with CRISPR-Cas9 genome editing. Exact coordinates and sequences of target sites, deletions, and genotyping primer sequences can be found in Supplementary Data 5. **B–G** Brachyury/T antibody staining (red) of E9.5 embryos. White dashed square in panels represents location of right bottom inserts with 2x magnification. Brachyury/T protein expression in the notochord is dose-dependent on the three enhancer elements. Asterisks in (**D–G**) mark absent notochord in rostral portion of the embryo. Scale bar in (**B**): 1 mm, applies to panels (**B–G**). **H–M** Overall morphology of E12.5 embryos with different genotypes. Blue lines indicate the location of immunofluorescence and H&E sections. Spina bifida and tail defects are dose-dependent. Arrowheads mark rudimentary tails. White lines mark spina bifida. Scale bar in **H**: 1 mm, applies to (**H–M**). **N–S** Dorsal view of embryos (sectioned at blue line in **H–M**). White lines mark areas of spina bifida. Arrowheads mark rudimentary tails compared to tails in wildtype control and double knock-out allele. Scale bar in (**N**): 2.5 mm, applies to panels (**N–S**). **T–Y** Immunofluorescence of mouse transverse sections. Anti-Sox2 labels the neural plate, anti-Tbxt the notochord, and DAPI marks nuclei. Sox2 expression is comparable amongst all genotypes, even in the genotypes with spina bifida, while there is loss of Brachyury/T staining in the notochord with increased loss of the enhancers. Arrowheads point to notochord. Asterisks mark absent notochord. Scale bar in (**T**): 0.2 mm, applies to panels (**T–Y**). **Z–E'** H&E staining of transverse sections confirm the dose-dependent loss of the notochord and spina bifida. Arrowheads point to notochord. Asterisks mark absent notochord. Scale bar in (**Z**): 0.2 mm, applies to (**Z–E'**).

correlate with gene dosage[54]. Importantly, the $T^{\Delta T3/\Delta T3,C,I}$ genotype with severely reduced Brachyury/T/Tbxtb protein levels is consistent with the loss of Brachyury/T/Tbxtb protein in the notochord in mice trans-heterozygous for the *TNE* deletion and a large, locus-spanning *Brachyury/T/Tbxtb* deletion that includes elements *C* and *I*[27], revealing the actual relevant enhancer regions (Figs. 1, 3, and 4) and motifs (Fig. 2). Finally, E9.5 homozygous triple knockout *ΔT3,C,I* embryos ($T^{\Delta T3,C,I/\Delta T3,C,I}$) showed a complete absence of Brachyury/T/Tbxtb protein in the entire notochord region ($n = 5/5$) yet all embryos retained Brachyury/T/Tbxtb protein in the tailbud ($n = 5/5$) (Fig. 5G). Taken together, our data establish the notochord-specific *Brachyury/T/Tbxtb* loss-of-function mutant in mice by means of deleting three conserved enhancer elements *in cis*.

Next, we examined phenotypic defects resulting from perturbed *Brachyury/T/Tbxtb* expression using various allele combinations involving *ΔC,I and ΔT3,C,I*. Consistent with the phenotypes at E9.5 (Fig. 5B–G), we observed a gradual increase of phenotype severity with deletion of the three different enhancer elements at E12.5 (Fig. 5H–E'). Wildtype control ($n = 25/25$) (Fig. 5H, N), homozygous $T^{\Delta C,I/\Delta C,I}$ embryos ($n = 24/24$) (Fig. 5I, O), heterozygous $T^{+/\Delta C,I}$ ($n = 5/5$), heterozygous $T^{+/\Delta T3}$ ($n = 23/23$) and $T^{+/\Delta T3,C,I}$ embryos ($n = 23/23$) (Supplementary Fig. 5G–I) appeared grossly normal. In contrast, we observed rudimentary tails with additional enhancer deletions. Rudimentary tails appeared in trans-heterozygous $T^{\Delta C,I/\Delta T3,C,I}$ embryos in 4.7 % ($n = 2/43$) (Fig. 5J, P) and were fully penetrant in homozygous $T^{\Delta T3/\Delta T3}$ ($n = 12/12$) (Fig. 5K, Q) similar to homozygous *TNE* embryos[27], and trans-heterozygous $T^{\Delta T3/\Delta T3,C,I}$ embryos ($n = 14/14$) (Fig. 5L, R), as well as in triple homozygous $T^{\Delta T3,C,I/\Delta T3,C,I}$ embryos ($n = 18/18$) (Fig. 5M, S). In addition, homozygous $T^{\Delta T3/\Delta T3}$ embryos ($n = 11/12$) (Fig. 5Q) seemed to display defects in neural tube closure very close to the tail, comparable to spina bifida; upon sectioning however, we identified this region to be very small and not a fully developed spina bifida phenotype (Fig. 5Q). In comparison, trans-heterozygous $T^{\Delta T3/\Delta T3,C,I}$ embryos displayed caudal spina bifida with 100% penetrance ($n = 14/14$) (Fig. 5R). Finally, triple-homozygous $T^{\Delta T3,C,I/\Delta T3,C,I}$ embryos lacking all three enhancers displayed spina bifida along 3/4 of the spine ($n = 18/18$) (Fig. 5S), reminiscent of previous observations using *Brachyury/T/Tbxtb*-targeting RNAi in mouse embryos[55,56]. These results provide compelling phenotypic evidence of the impact of cumulative enhancer deletions on *Brachyury/T/Tbxtb* expression in the notochord.

We further validated these phenotypes with immunohistochemistry and histology. We visualized Brachyury/T/Tbxtb protein in transversal sections of E12.5 embryos together with the neural plate marker Sox2: compared to wildtype (Fig. 5T), heterozygous $T^{+/\Delta C,I}$, $T^{+/\Delta T3}$, $T^{+/\Delta T3,C,I}$ (Supplementary Fig. 5J–L) as well as homozygous $T^{\Delta C,I/\Delta C,I}$ (Fig. 5U) embryos that were all grossly normal, we found decreased Brachyury protein in the notochord of $T^{\Delta C,I/\Delta T3,C,I}$ (Fig. 5V) and $T^{\Delta T3/\Delta T3}$ (Fig. 5W) embryos. Strikingly, we observed a complete absence of Brachyury protein in $T^{\Delta T3/\Delta T3,C,I}$ embryos (Fig. 5X) and $T^{\Delta T3,C,I/\Delta T3,C,I}$ (Fig. 5Y) embryos. In contrast, Sox2 expression was comparable in all embryos (Fig. 5T–Y, Supplementary Fig. 5J–L), even in $T^{\Delta T3,C,I/\Delta T3,C,I}$ embryos that clearly displayed spina bifida along the entire spine (Fig. 5Y). Compared to wildtype embryos (Fig. 5Z), additional histology assessed by H&E staining confirmed wildtype-looking notochords in $T^{+/\Delta C,I}$, $T^{+/\Delta T3}$, $T^{+/\Delta T3,C,I}$, and homozygous $T^{\Delta C,I/\Delta C,I}$ embryos (Supplementary Fig. 5M–O, Fig. 5A'), smaller (in diameter) notochords in $T^{\Delta C,I/\Delta T3,C,I}$ (Fig. 5B') and $T^{\Delta T3/\Delta T3}$ (Fig. 5C') embryos, and absent notochords in $T^{\Delta T3/\Delta T3,C,I}$ and $T^{\Delta T3,C,I/\Delta T3,C,I}$ embryos (Fig. 5D'–E').

We found that the two most severe enhancer mutants are not viable as adults since we did not recover homozygous triple $T^{\Delta T3,C,I/\Delta T3,C,I}$ ($n = 0/59$) or trans-heterozygote $T^{\Delta T3/\Delta T3,C,I}$ ($n = 0/31$) animals at term (Supplementary Fig. 5P), indicating lethality prior to or shortly after birth. In contrast, homozygous $T^{\Delta T3/\Delta T3}$ animals were born, but died within 14 days after birth, with one exception where we identified one homozygous $T^{\Delta T3/\Delta T3}$ ($n = 1/34$) animal without a tail that survived until adulthood (Supplementary Fig. 5P). In contrast, $T^{\Delta C,I/\Delta T3,C,I}$ ($n = 46$) trans-heterozygotes and homozygous $T^{\Delta C,I/\Delta C,I}$ ($n = 100$) animals survived to adulthood (Supplementary Fig. 5P). Notably, a variable percentage of $T^{\Delta C,I/\Delta C,I}$, $T^{\Delta C,I/\Delta T3,C,I}$, and $T^{+/\Delta T3}$ animals presented with kinked tails (Supplementary Fig. 5Q), with two $T^{\Delta C,I/\Delta T3,C,I}$ animals displaying a small tail (Supplementary Fig. 5R), reminiscent of hypomorphic *Brachyury/T/Tbxtb* mutants and in vivo *Brachyury/T/Tbxtb* knockdown by siRNA[9,27,55,56]. Taken together, our data are consistent with the correlation of *Brachyury/T/Tbxtb*-mutant phenotypes and gene dosage controlled by enhancer activity, as revealed by increasing phenotype severity with an increasing number of combined enhancer deletions in *Brachyury/T/Tbxt*.

In summary, our data establishes that the combined activity of the enhancers *T3*, *C*, and *I* in the mouse *Brachyury/T/Tbxtb* locus are necessary to convey notochord expression of *Brachyury/T/Tbxtb*. Upon combined loss of these enhancers, the notochord is lost.

### *T3, C* and *I* are conserved among jawed vertebrates

The evolutionary trajectory of chordate *Brachyury* control in the notochord remains unresolved. The notochord-regulatory elements driving *Brachyury* expression in Ciona are promoter-proximal[8,10,31]. Zebrafish *tbxta/ntla* harbors a −2.1 kb upstream notochord element containing the two smaller elements *E1* and *E2*[23]. In contrast, zebrafish *tbxtb* descended from the same ancestral *Brachyury* gene as the single mammalian *Tbxtb* gene. Further, while zebrafish *tbxtb* remains expressed in the notochord[21,57], its regulatory elements have not been reported. Using direct sequence comparisons of mammalian *T3, C,* and *I* to the zebrafish genome, we did not find any sequences of significant sequence similarity (Fig. 1A).

Identifying non-coding sequence conservation across vertebrate lineages, whether from human or other tetrapods to the fast-evolving teleost fishes like zebrafish, remains notoriously challenging. Species with slow rates of molecular evolution can help as "genomic bridges" to provide sequence connectivity across distant vertebrate groups[58,59]. The spotted gar (*Lepisosteus oculatus*) is a slowly evolving ray-finned fish that has diverged from zebrafish and other teleosts before a teleost-specific whole-genome duplication, providing a bridge species

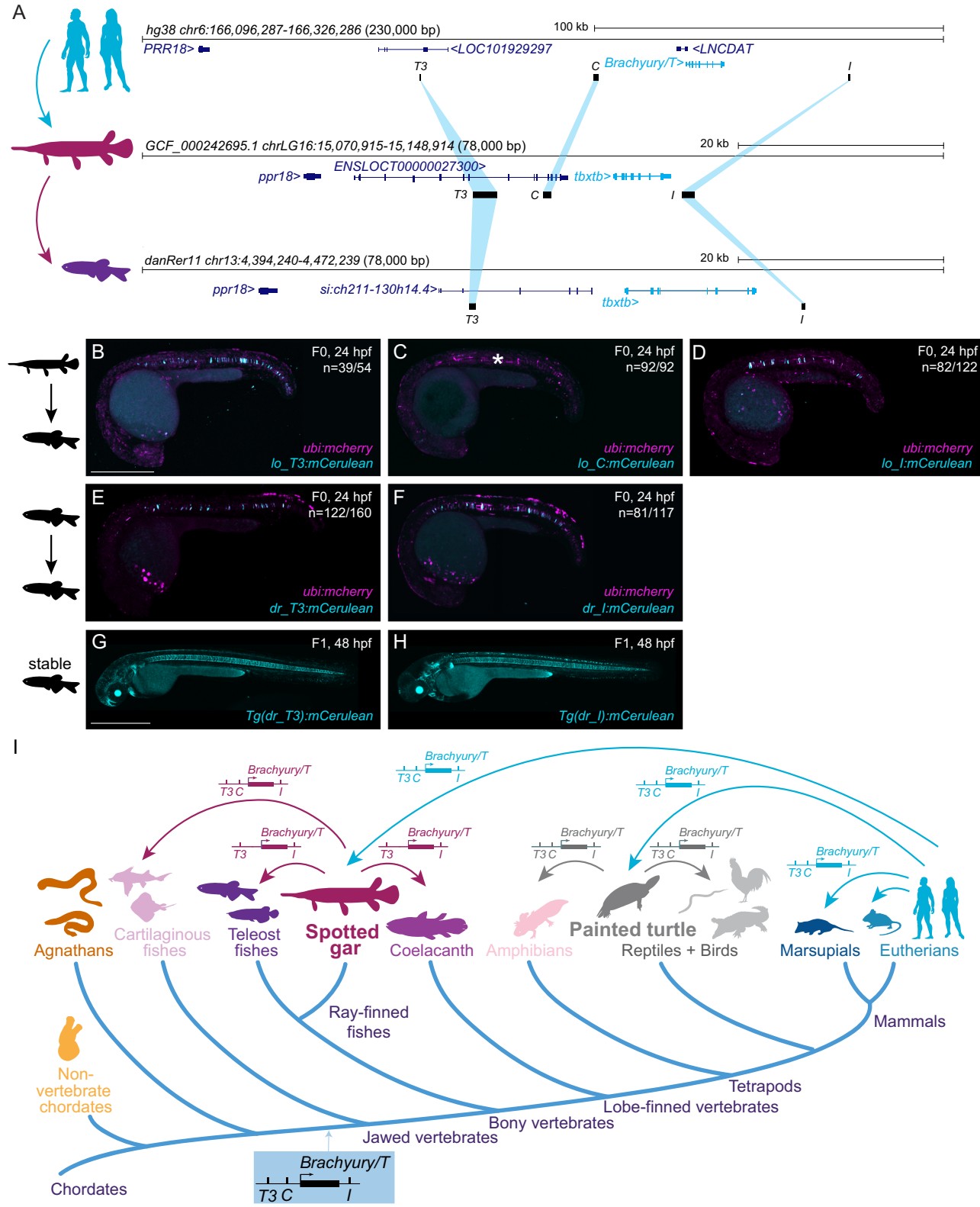

for genomic comparisons between tetrapods and teleosts[58]. Using BLAST searches, we found sequence similarity between human *T3, C,* and *I* and regions of the spotted gar *tbxtb* locus with equivalent positions relative to the gar *tbxtb* gene body compared to mammals (Fig. 6A). Next, we used these spotted gar *T3, C,* and *I* regions as BLAST queries to bridge to the genomes of zebrafish and other fish lineages (Supplementary Data 4). This approach uncovered candidate regions for *T3* and *I*, but not *C*, within the zebrafish *tbxtb* locus (Fig. 6A).

Analogous to our tests with mammalian enhancer candidates, we cloned reporter transgenes coupled with the *betaE-globin:mCerulean* cassette using the *T3, C,* and *I* enhancer elements from the spotted gar *tbxtb* locus. Upon injection into zebrafish embryos, both spotted gar *lo_T3* and *lo_I* displayed specific and reproducible notochord reporter activity (*n* = 39/54, *n* = 82/122) (Fig. 6B, D, Supplementary Data 2). In contrast, and akin to the mouse *mm_C* enhancer element, spotted gar element *lo_C* did not convey any notochord reporter activity in

**Fig. 6 | Bridge species establish the presence of *Tbxtb* enhancers across jawed vertebrates. A** Location of the enhancer elements in the human (top), gar (middle), and zebrafish (bottom) *Brachyury/T/Tbxtb* loci, adapted from the UCSC browser as established through the "gar bridge". **B–D** Representative F0 zebrafish embryos injected with the gar enhancer elements *lo_T3* (**B**), *lo_C* (**C**), and *lo_I* (**D**). *T3* and *I* show mosaic *mCerulean* reporter expression in the notochord at 24 hpf compared to gar element *C* with is not active in the zebrafish notochord (asterisk). N represent the number of animals expressing mCerulean in the notochord relative to the total number of animals expressing mosaic *ubi:mCherry* as injection control. Scale bar in (**B**): 0.5 mm, applies to (**B–F**). **E, F** Representative F0 zebrafish embryos injected with the conserved zebrafish enhancer elements *dr_T3* (**E**) and *dr_I* (**F**). *T3* and *I* show mosaic *mCerulean* reporter expression in the notochord at 24 hpf. N represent the

number of animals expressing mCerulean in the notochord relative to the total number of animals expressing mosaic *ubi:mCherry* as injection control.
**G, H** Representative images of stable F1 embryos at 2 dpf of zebrafish enhancer elements *T3* and *I* recapitulate the F0 expression pattern in the notochord, with *dr_T3* (**E**) additionally expressing mCerulean in the brain, heart, and fin, and *dr_I* (**G**) in the proximal kidney close to the anal pore, pharyngeal arches, heart, fin, and spinal cord neurons. Scale bar in (**G**): 0.5 mm, applies to (**G, H**). **I** Phylogenetic representation of species investigated using the bridging approach with spotted gar and painted turtle as anchor species within ray-finned fish and tetrapod lineages. Arrows indicate informative phylogenetic comparisons to uncover conservation of enhancer elements *T3*, *I*, and *C*. The species silhouettes were adapted from the PhyloPic database (www.phylopic.org).

zebrafish embryos (*n* = 0/92) (Fig. 6C, Supplementary Data 2). The zebrafish-derived *dr_T3* and *dr_I* also showed selective notochord activity when tested in zebrafish transgenic reporter assays (*n* = 122/160, *n* = 81/117) (Fig. 6E, F, Supplementary Data 2). Further confirming our results, we found robust reporter activity in the notochord of stable transgenic zebrafish lines based on *dr_T3* and *dr_I* (Fig. 6G, H). All fish enhancer elements started to express the *mCerulean* reporter during early somitogenesis, similar to the human elements.

Using the three gar elements as queries, in addition to clupeocephalan teleosts (e.g. zebrafish), we found *T3* and *I* also in the other two major teleost lineages elopomorphs (e.g. eel) and osteoglossomorphs (e.g. arowana). However, we did not detect any equivalent sequence for *C* in any teleosts, indicating that this element has been lost or diverged beyond recognition in the teleost lineage (Fig. 6I). However, we detected orthologs of all three elements, including *C*, at expected locations around the *tbxtb* genes in additional non-teleost ray-finned fishes (e.g. bowfin, sturgeon, reedfish) as well as in the more basally diverging cartilaginous fishes (e.g. sharks, skate) (Supplementary Data 4); in contrast, we only detected *T3* and *I* in the lobe-finned coelacanth (Fig. 6I). To explore the presence of the three enhancer elements among tetrapods, we used the painted turtle, characterized by a slow genomic evolutionary rate[60,61], as an additional bridge species within tetrapods. We found all three elements in the turtle *Brachyury/T/Tbxtb* locus and through use of the painted turtle as reference also in other reptiles and birds, as well as in amphibians (e.g. axolotl) (Fig. 6I, Supplementary Data 4), but did not detect any of the three elements in the jawless cyclostome (e.g. lamprey, hagfish) genomes. Furthermore, we found that the human T-box motifs, which we identified using FIMO (Fig. 2) in our enhancers, are conserved across tetrapods and fishes as distantly related as ghost shark based on sequence alignments (Supplementary Fig. 6A–C) as well as multi-species FIMO analyses (Supplementary Data 7). Cross-species sequence conservation centers at the T-box motifs (Supplementary Fig. 6A–C) which supports both their functional importance as well as their evolutionary ancestry since at least the last common ancestor of jawed vertebrates.

Taken together, our observations provide strong evidence that notochord enhancers *T3*, *I*, and *C* are deeply conserved *cis*-regulatory elements of the *Brachyury/T/Tbxtb* gene that were already present in the last common ancestor of jawed vertebrates over 430 million years ago.

## Discussion

How the *Brachyury/T/Tbxtb* gene is controlled during notochord development is fundamental to our understanding of how basic concepts of body plan formation remain conserved or have diverged across species. Shadow enhancers, seemingly redundant transcriptional *cis*-regulatory elements that regulate the same gene and drive overlapping expression patterns, are a pervasive feature of developmental gene regulation[62]. The concept of enhancer redundancy through one or more shadow enhancers acting on the same gene in addition to a primary enhancer has been established for numerous loci[62–67]. Shadow enhancers are thought to provide robustness to gene

expression and buffer against genetic and environmental variations[62,65], a hypothesis validated in mammals[66,67].

Here, we discovered a deeply conserved set of three notochord-specific shadow enhancers within the human *TBXT* locus as ancient *cis*-regulatory elements. While we cannot draw conclusions about reporter initiation or early reporter expression patterns, cross-species enhancer testing reveals that the *cis*-regulatory grammar of the three human enhancers *T3*, *C*, and *I*, is correctly interpreted in vertebrates including mice, salamanders, and zebrafish, but not in the invertebrate chordate Ciona. The three notochord enhancers described here are not the only non-coding conserved elements across mammalian *Brachyury/T/Tbxtb* loci (Figs. 1A, 3A, and 4A). Even though our zebrafish reporter assays did not reveal any notochord activity in three out of the six tested human enhancer elements (*K*, *J*, and *L*), we cannot rule out synergistic or interdependent notochord activity conveyed by additional elements. Further, our reporter assays indicate that not all three *Brachyury/T/Tbxtb* notochord enhancers *T3*, *C*, and *I* have equal potency. Enhancer element *C* shows variable activity and remains unrecognized in teleost fishes and Coelacanth. Compared to human *C* with reproducible notochord activity in all tested models (Fig. 1C, F, I, M) and *Monodelphis C* that is active in zebrafish and uniquely in Ciona (Fig. 4C, E), mouse *C* showed no discernible activity in any assay including in mouse embryos (Fig. 3C, G) despite significant sequence conservation. We speculate that while mouse *C* is not active in isolation, it may contribute together with *T3* and *I* to *Brachyury* activity in the notochord. This model is consistent with the impact of *TNE* deletions when combined with larger deletions that include *TNE* and *C* in mouse trans-heterozygotes[27] (Fig. 5). The potential auto-regulation of *Brachyury/T/Tbxtb* by its protein product via in part conserved T-box motifs in enhancers *T3* and *I* might contribute to the enhancer redundancy and divergent activity of element *C* when tested in isolation (Fig. 2). Our data propose that enhancer *C* is an auxiliary element to *T3* and might contribute to duration, expression levels, or other features that differ among *Brachyury/T/Tbxtb* notochord expression across vertebrates. Our combined data proposes a model in which notochord expression of vertebrate *Brachyury/T/Tbxtb* is cumulatively or cooperatively driven by enhancers *T3*, *C*, and *I*. In this model, sequence variants of *T3*, *C*, and *I* that modulate their individual potency became selected for modulating Brachyury/T levels to species-specific requirements.

The conservation of gene order (micro-synteny) between species can be indicative of the presence of *cis*-regulatory elements, which are crucial for controlling expression of the physically linked genes[68]. The finding of functionally relevant distant enhancers 5' and 3' of the *Brachyury/T/Tbxtb* gene body is further supported by the conserved gene linkage *Sftd2-(Prr18)-Brachyury/T/Tbxtb-Pde10a* across the entire jawed vertebrate phylogeny. In agreement with a distinct gene linkage surrounding *Brachyury/T/Tbxtb* in agnathans (Fig. 6I), we were unable to identify any of the three distant enhancers in two species representing this clade. Likewise, a distinct gene linkage associates with *Tbxta*, the second *Tbxtb* paralog in fish, which apparently lacks any of the three notochord enhancers described here. *tbxta/ntla* expression is instead

controlled by two mesoderm/notochord enhancers located close to the gene promoter (Harvey et al., 2010), a possible example of evolutionary novelty following ancestral gene duplication. In contrast, the functionally less impactful zebrafish *tbxtb/ntlb* gene retained the regulation of the *Tbxtb* gene from the jawed vertebrate ancestor (Fig. 6). We did not find any evidence for sequence conservation of the *Tbxtb* *T3, I*, or *C* regions within vertebrate *Tbxta* loci or any other genomic regions. Future detailed studies across vertebrate *Tbxt* paralogs are needed to evaluate whether or not the three *Tbxtb* regulatory elements identified here were already part of the single *Tbxt* gene in a vertebrate ancestor. Notably, zebrafish mutants of *tbxta/ntla* have been widely studied as model for *Brachyury* function in notochord formation[13,15,69], while the seemingly less impactful *tbxtb* has retained ancestral regulation. Why zebrafish, and possibly other fish lineages, use *tbxta* as their main functional Brachyury paralog, and how the regulatory balance between *T3, C*, and *I* plays out across individual vertebrate lineages, warrants future efforts.

We found that *Brachyury/T/Tbxtb* notochord enhancers *T3* and *I*, and possibly further supported by enhancer *C*, represent a shadow enhancer combination that contributes to the robust *Brachyury/T/Tbxt* expression in mammals. In mice, neither deletion of enhancer *T3/TNE*[27], nor deletion of enhancer *C, I*, or C and I, resulted in a discernable notochord phenotype (Fig. 5). Nonetheless, by combining deletions of all three notochord enhancer elements, we showed a dose response for Brachyury/T expression in the notochord. In particular, in embryos where *ΔT3* is combined with a chromosome harboring *ΔT3,C,I* as trans-heterozygotes ($T^{\Delta T3/\Delta T3,C,I}$) or in triple homozygous knock-out embryos ($T^{\Delta T3,C,I/\Delta T3,C,I}$), we observed loss of Brachyury/T protein in the notochord as well as notochord-specific phenotypes, such as spina bifida (Fig. 5). The neural tube closure defects are similar to phenotypes observed in *Brachyury/T/Tbxtb* knockdown embryos[55,56] or hypomorphic *Brachyury/T/Tbxtb* mutants[9]. These results assign an essential, combinatorial role to the enhancer elements *T3/TNE, C* and *I* in regulating *Brachyury/T/Tbxtb* in the notochord. Notably, previous work[70,71] has described the *T2* mutant caused by a large viral integration 5′ of the mouse *Brachyury/Tbxt* locus that (i) is recessive lethal with phenotypes reminiscent of *Brachyury* loss, and (ii) does complement loss-of-function alleles for Brachyury. *T2* has been hypothesized to encode a short protein off a long mRNA[70,71]. The described genomic position of the viral integration in *T2* places it in the vicinity and upstream of enhancer element *C*. We note that various vertebrate *Brachyury/tbxtb* loci feature annotated long non-coding RNAs upstream of the main gene body that are reminiscent of enhancer RNAs (Figs. 3A and 6A). We therefore hypothesize that the *T2* mutation is caused by a disruption of the gene-regulatory landscape of the mouse *Brachyury/Tbxt* gene by the viral integration, changing the interaction of distant enhancer elements with the promoter. Inspection of the chromatin landscape of the *Brachyury/Tbxt* locus, also in *T2* mutants, could shed light on the architecture of the locus during notochord development.

The significance of *Brachyury/T/Tbxtb* regulation in the notochord translates to chordoma tumors that feature expression of this T-box transcription factor as key diagnostic readout[32,72,73]. Both sporadic and familial chordoma are hypothesized to derive from notochord remnants in the spine that do not convert to nucleus pulposus tissue[32,74,75]. Native Brachyury-expressing cells in the nucleus pulposus decrease in number with age along with a concomitant increase in cartilage-like cells[4,76–78]. What role these long-lasting Brachyury-positive cells play in the adult spine, if they progressively differentiate into cartilage, and how *Brachyury* gene activity is sustained, remains unknown. Detection of Brachyury protein is a diagnostic marker for chordoma[32], yet the functional contribution of its re-activated or persistent expression in the tumor is not known[56,79–81]. Our analysis of reported familial and sporadic chordoma amplifications indicate that amplifications invariantly retain the notochord enhancer *I* together with the gene body

including the promoter[34,37]. Enhancer *I* lies within a super-enhancer region identified in chordoma cell lines[38], further implicating its transcriptional engagement in chordoma. Amplifications occurring in tandem with the original locus propose a scenario where the retained enhancer *I* could synergize with *C* and *T3* from the original locus on the newly amplified gene copies, potentially resulting in increased *Brachyury/T/TBXTB* expression (Fig. 1A). Beyond chordoma, changes in enhancer sequence or relative distance to the *Brachyury/T/TBXTB* gene body could also impact spine formation and health by altering the robustness of Brachyury expression in the notochord and subsequent nucleus pulposus.

Tremendous progress with in vitro differentiation regimens have resulted in stem cell-derived models for body segmentation and different organ structures[82–85]. However, notochord formation has only been reported in more complex systems that recapitulate major hallmarks of embryo patterning[86–88]. Reporters based on our isolated enhancers could potentially provide potent readouts to screen for differentiation regimens that result in notochord fates. Together, our uncovered set of shadow enhancers in *Brachyury/T/TBXTB* advance our concepts of how this key contributor to notochord formation is regulated and de-regulated in development and disease.

## Methods
### Ethical regulations
All research within this manuscript complies with all relevant ethical regulations that are described and named individually in each paragraph.

### *Brachyury* locus annotations
The UCSC genome browser was used to identify and visualize enhancer elements in the human, mouse, and Monodelphis *Brachyury* locus. *.bed files were generated with the approximate genomic location of human *Brachyury* amplifications in chordoma tumors from different patients[34,37]. Previously published ATAC-sequencing data of U-CH2 cells and MUGCHOR cells[38], as well as Brachyury/T ChIP sequencing data of human embryonic stem cells (hESCs)[40] and U-CH1 cells[36] were added. Further, the repeat masker track, ENCODE cCREs, layered H3K27ac, and the conservation track for mouse and Monodelphis were added. Ultimately, using this strategy, the human enhancer element candidates *T3, K, J, C, I*, and *L* were identified. For detailed information, see Supplementary Data 1 and 3.

The same strategy was applied to find the corresponding mouse enhancer elements. Published ATAC-seq data of mouse ESCs[89] and Brachyury/T-positive fluorescence-activated cell sorted cells from the caudal ends of wild-type mouse embryos (TS12/8 dpc and TS13/8.5 dpc)[90], as well as Brachyury/T ChIP sequencing data of mouse ESCs[39,90] were used. Again, the repeat masker track, the ENCODE Candidate Cis-Regulatory Elements (cCREs, combined from all cell types) track, tracks containing H3K27ac, H3K4me, DNase signals from E11.5 neural tube as it likely contains notochord tissue as well due to extraction[91], and the Vertebrate Multiz Alignment & Conservation track to check for conservation in human, Monodelphis, and zebrafish, were added. This approach identified the mouse enhancer element candidates *T1, T2, T3, J, C2/next to C, C, Tstreak, I, T4, T5*, and *T6*, of which *T1, T3, J, C, Tstreak, I, and T5* were pursued and tested (Supplementary Data 3 and 5).

To find the corresponding *Monodelphis* elements, the repeat masker and 9-Way Multiz Alignment & Conservation track were included to identify *T3, C*, and *I* (Supplementary Data 3 and 5).

### Cloning of the enhancer element reporter plasmids
Each *Brachyury* enhancer element candidate was amplified from either human, mouse, Monodelphis, spotted gar, or zebrafish genomic DNA using the Expand Hi-Fidelity PCR System (11732641001, Roche). Exact coordinates are listed in Supplementary Data 3. Each enhancer

candidate was TOPO-cloned into the *pENTR5′-TOPO* plasmid (K59120, Invitrogen) according to the manufacturer's instructions (half-volume reactions). Subsequent Multisite Gateway cloning were performed using LR Clonase II Plus (12538120, Invitrogen) according to the manufacturer's instructions (half-volume reactions) and recommended reaction calculations[92]. 5′ entry plasmids containing the different enhancer elements were assembled into reporter expression plasmids together with the middle entry plasmid (*pME*) containing the mouse *betaE-globin* minimal promoter expressing mCerulean (*pSN001*) as well as mApple (*pCK068*), the 3′plasmid #302 (*p3E_SV40polyA*), and the destination plasmid pDESTTol2A2 containing *crybb1:mKate2* (*pCB59*) and *pDESTexorh:EGFP* containing EGFP expression in the pineal gland (*pCK017*) as transgenesis markers[42]. Assembled vectors were verified using restriction digest and Sanger sequencing using standard sequencing primers for Multisite Gateway assemblies[42,92].

## Zebrafish husbandry, transgenic reporter assays and stable transgenic lines

Zebrafish animal care and procedures were carried out in accordance with the IACUC of the University of Colorado Anschutz Medical Campus (protocol # 00979), Aurora, Colorado. Adult AB and TU wildtype zebrafish were obtained from the Zebrafish International Resource Center (ZIRC) and maintained as per standard husbandry procedures[93].

To test the transient activity of the putative enhancer elements, 25 ng/μL *Tol2* mRNA, 12.5 ng/μL reporter expression plasmid DNA, and 12.5 ng/μL *ubi:mCherry* plasmid[94] as injection control were co-injected into one-cell stage wild-type zebrafish embryos[44]. At 24 hpf, embryos were anesthetized with 0.016% Tricaine-S (MS-222, Pentair Aquatic Ecosystems Inc.) in E3 embryo medium and embedded in E3 with 1% low-melting-point agarose (A9045, Sigma Aldrich).

To generate stable transgenic lines, 25 ng/μL *Tol2* mRNA were co-injected with 25 ng/μL reporter expression plasmid DNA[95,96]. Multiple F0 founders were screened for specific *mCerulean* and *mKate2* expression, raised to adulthood, and screened for germline transmission. Resulting F1 single-insertion transgenic strains were established and verified through screening for a 50% germline transmission rate outcrosses in the subsequent generations as per our previously outlined procedures[96]. *Tg(drl:mCherry)* was used as a marker for lateral plate mesoderm derivatives[41].

For imaging, embryos were mounted laterally on glass bottom culture dishes (627861, Greiner Bio-One) and confocal images were acquired with a Zeiss LSM880 using a ×10/0.8 air-objective lens. Fluorescence channels were acquired sequentially with maximum speed in bidirectional mode in 3 μM slices. The range of detection for each channel was adapted to avoid any crosstalk between the channels. Images of acquired Z-stacks were reconstructed with ImageJ/Fiji as a maximum intensity projections.

## Axolotl husbandry, transgenic reporter assays and immunostaining

Procedures for care and manipulation of all animals used in this study were performed in compliance with the laws and regulations of the State of Saxony, Germany. Axolotl husbandry and experiments (non-free feeding stages) were performed at the Center for Regenerative Therapies Dresden (CRTD), Dresden, Germany. Adult axolotls (*Ambystoma mexicanum*) were obtained from the axolotl facility at the Technische Universität Dresden (TUD)/CRTD Center for Regenerative Therapies Dresden. Animals were maintained in individual aquaria at -18–20 °C[97]. Axolotls of the white (d/d) strain were used in all experiments.

Transgenic axolotl embryos were generated using *Tol2* transposase following standard protocols[98]. For live imaging, the embryos were anaesthetized by bathing in 0.01% benzocaine and imaged on an Olympus SZX16 fluorescence stereomicroscope. Embryos were staged as described previously[99].

For immunostaining, axolotl embryos were fixed in MEMFA at 4 °C overnight, washed in PBS, embedded in 2% low-melting temperature agarose, and sectioned by vibratome into 200 μm-thick sections. Fibronectin was detected using mouse anti-Fibronectin (ab6328, Abcam; dilution 1:400) and donkey anti-mouse Alexa Fluor™ 568 (A-10037, Invitrogen; dilution 1:600). After staining, sections were mounted with Mowiol (81381, Millipore Sigma). Confocal images were acquired on a Zeiss LSM780-FCS inverted microscope.

## Transgenic mouse reporter assays

Research was conducted at the E.O. Lawrence Berkeley National Laboratory (LBNL) and performed under U.S. Department of Energy Contract DE-AC02-05CH11231, University of California (UC). Transgenic mouse assays were performed in *Mus musculus* FVB mice (obtained from The Jackson Laboratory), animal protocol number 290003; reviewed and approved by the Animal Welfare and Research Committee at Lawrence Berkeley National Laboratory.

For comprehensive analysis of species-specific *T3*, *C* and *I*, enSERT enhancer analysis was used, allowing for site-directed insertion of transgenic constructs at the *H11* safe-harbor locus[100,101]. EnSERT is based on co-injection of Cas9 protein and *H11*-targeted sgRNA in the pronucleus of FVB single cell-stage mouse embryos (E0.5) with the targeting vector encoding a candidate enhancer element upstream of the *Shh*-promoter-*LacZ* reporter cassette[45]. Enhancer elements were PCR-amplified from human, mouse and Monodelphis genomic DNA and cloned into the respective *LacZ* expression vector[102]. Embryos were excluded from further analysis if they did not contain a reporter transgene in tandem. CD-1 females (The Jackson Laboratory) served as pseudo-pregnant recipients for embryo transfer to produce transgenic embryos which were collected at E9.5 and stained with X-gal using standard methods[102].

## Histological analysis of Nuclear Fast Red-stained sections from transgenic mouse embryos

After LacZ staining, E9.5 transgenic mouse embryos were dehydrated in serial alcohols (1 × 70%, 1 × 80%, 1 × 90%, 2 × 96%, 2 × 100% ethanol, followed by 1 × 100% isopropanol for 20 min each) and cleared twice for 30 min with Histo-Clear II (HS-202, National Diagnostics) for paraffin wax embedding. 10 μm-thick transverse sections were obtained with a Leica Biosystems RM2245 Semi-Automated Rotary Microtome. Sections were de-waxed, rehydrated, and stained with Nuclear Fast Red (R5463200, Ricca Chemical) for 2 min. After staining, sections were dehydrated and mounted with Omnimount (HS-110, National Diagnostics). Images were obtained using a Leica M205 FA stereo microscope.

## Ciona reporter assays

Ciona experiments were performed at UCSD as described previously[29,103]. Adult *Ciona intestinalis* type A aka *Ciona robusta* (obtained from M-Rep) were maintained under constant illumination in seawater (obtained from Reliant Aquariums) at 18 °C. Briefly, human, mouse and Monodelphis enhancer elements *T3*, *C* and *I* were subcloned into appropriate plasmids suited for expression in Ciona, upstream of a basal Ciona Forkhead promoter driving GFP[28,104]. Ciona embryos were electroporated with 70 μg of each plasmid as previously described[105] and reporter expression was counted blind in 50 embryos per biological repeat. All constructs were electroporated in three biological replicates. Images were taken of representative embryos with an Olympus FV3000 microscope using a 40X objective.

## Deletion of mouse enhancer elements *T3, C,* and *I*

All mouse experimental procedures and animal care were approved by the Animal Care Committee of the Institute of Molecular Genetics

(IMG), Czech Academy of Sciences, Prague, Czech Republic, and covered under protocol permission number 357/2021. Experiments were performed in compliance with the European Communities Council Directive of November 24, 1986 (86/609/EEC), as well as national and institutional guidelines.

For this study, inbred C57BL/6 N mice (The Jackson Laboratory) were used. Mice carrying deletions of enhancer elements *T3*, *C*, and *I* were generated using CRISPR-Cas9 technology. The cRNAs (purchased from Integrated DNA technologies, IDT) were designed to target the 5' and 3' ends of the mouse enhancer elements *T3*, *C* and *I* to delete the genomic regions in between. For genomic location and sequence of the selected target sites, as well as genomic coordinates of the deleted enhancer element sequences, see Supplementary Data 5.

A ribonucleoprotein (RNP) complex of crRNA/TRACR (1072532, IDT) and SpCas9 protein (1081058, IDT) was electroporated into fertilized zygotes isolated from C57BL/6 N mice. Zygote electroporation and transfer into pseudo-pregnant foster females was performed as previously described[106]. Founder animals from multiple embryo transfers were genotyped from tail biopsies using PCR and Sanger sequencing and the positive animals were backcrossed to C57BL/6 N mice.

Independent knockout lines for enhancer element $C$ ($\Delta C$) and $I$ ($\Delta I$) were generated. Heterozygous $\Delta C$ and $\Delta I$ ($T^{+/\Delta C}$ and $T^{+/\Delta I}$) and homozygous $\Delta C$ and $\Delta I$ ($T^{\Delta C/\Delta C}$ and $T^{\Delta I/\Delta I}$) embryos were investigated for potential overall phenotypes, but appeared phenotypically normal. Pups were born normally and grew up into fertile adults.

To generate a double knockout $\Delta C,I$ strain, homozygous $T^{\Delta C/\Delta C}$ mice were used for electroporation of CRISPR-Cas9 RNP complexes deleting enhancer element *I*. Pups homozygous for $\Delta C,I$ ($T^{\Delta C,I/\Delta C,I}$) were born phenotypically normal and developed into fertile adults; however, around 20% of the animals had a kinked tail (Supplementary Fig. 5M, N).

To generate a triple knockout $\Delta T3,C,I$ mouse strain, heterozygous $\Delta C,I$ ($T^{+/\Delta C,I}$) mice were used for electroporation of CRISPR-Cas9 RNP complexes deleting enhancer element *T3* ($\Delta T3$). Heterozygous $T^{+/\Delta T3,C,I}$ or trans-heterozygous $T^{\Delta T3/\Delta C,I}$ embryos were phenotypically normal and grew up into fertile adults. To establish a single knockout line for enhancer element *T3* ($\Delta T3$), $T^{\Delta T3/\Delta C,I}$ animals were outcrossed to establish $T^{+/\Delta T3}$.

$T^{\Delta C,I/\Delta T3,C,I}$ animals were generated by mating $\Delta C,I$ ($T^{\Delta C,I/\Delta C,I}$) and $\Delta T3,C,I$ ($T^{+/\Delta T3,C,I}$) strains and $T^{\Delta T3/\Delta T3,C,I}$ by mating $\Delta T3$ ($T^{+/\Delta T3}$) and $\Delta T3,C,I$ ($T^{+/\Delta T3,C,I}$) strains, respectively. Finally, homozygous $T^{\Delta T3,C,I/\Delta T3,C,I}$ animals were generated by mating trans-heterozygous $\Delta C,I/\Delta T3,C,I$ ($T^{\Delta C,I/\Delta T3,C,I}$) animals.

Around 60% of $T^{\Delta C,I/\Delta T3,C,I}$ pups were born with a tail defect and adult animals displayed a kinked tail, with around 2% of the $T^{\Delta C,I/\Delta T3,C,I}$ pups displaying a small tail. In contrast, adult trans-heterozygous $T^{\Delta T3/\Delta T3,C,I}$ and homozygous $T^{\Delta T3,C,I/\Delta T3,C,I}$ animals were never recovered likely due to lethality at around birth or during early postnatal life.

The deletion breakpoints in the individual enhancer alleles were determined by Sanger sequencing. Mice were genotyped using PCR with dedicated primer sets (Supplementary Data 5). Mouse embryos of the given stage were harvested from timed pregnant mice. The day of plug was counted as embryonic day 0.5 (E0.5).

## E9.5 whole mount immunostaining and imaging
E9.5 mouse embryos were collected and whole mount immunostaining was done as previously described[107]. Brachyury/T/Tbxt expression in E9.5 embryos was visualized using rabbit anti-Brachyury (ab209665, Abcam; dilution 1:2000) and donkey anti-rabbit Alexa Fluor™ 594 (A-21207, Invitrogen, dilution 1:500). Images were obtained using a Zeiss AxioZoom V16 macroscope with Apotome with a Zeiss Axiocam 512 mono camera. A qualitative analysis of all investigated embryos can be found in Supplementary Data 6.

## E12.5 embryo preparation, immunostaining and imaging
E12.5 mouse embryos were collected and fixed overnight in 4% paraformaldehyde. Whole embryo images were acquired using a Olympus SZX9 stereo microscope with a Olympus DP72 camera. Afterwards, embryos were embedded in paraffin, and 9 μm-thick transverse sections were obtained using a Microtome Leica RM2255. Sections were deparaffinized, rehydrated, and stained with hematoxylin & eosin (H-3502, Vectorlabs) for histology, or rabbit anti-Brachyury (ab209665, Abcam; dilution 1:2000) and donkey anti-rabbit Alexa Fluor™ 594 (A-21207, Invitrogen, dilution 1:500), or goat anti-Sox2 Y-17 (sc-17320, Santa Cruz; dilution 1:400) and donkey anti-goat Alexa Fluor™ 488 (A-11055, Invitrogen, dilution 1:500) together with DAPI (10236276001, Roche Diagnostics) according to the manufacturer's instructions. After staining, sections were mounted with Mowiol (81381, Millipore Sigma). Images of sections were obtained using a Leica DM6000 widefield fluorescence microscope with a Leica DFC 9000 camera.

## Gar and turtle bridge alignment
To establish genomic connectivity across distant vertebrate lineages, a bridging approach that leverages species with slowly evolving genomic sequences, such as spotted gar within ray-finned fishes[58] and painted turtle within tetrapods[60], was used. Using human *T3*, *C*, and *I* as queries, BLASTN searches at ensembl.org[108] (search sensitivity: distant homologies) against the bridge species genomes were performed. Candidate BLAST hit regions were manually inspected for their location in relation to the *Tbxtb* gene locus for further consideration. Core regions based on the initial BLAST hits in both bridge species were expanded in both directions up to the next annotated repeat element. Once the three elements were established in the bridge species, their sequences were used for as queries for BLASTN searches with genomes representative species across all major vertebrate lineages as targets (see Supplementary Data 4 for species list, genome assemblies, and enhancer element coordinates). Further BLASTN chaining through additional species was performed as needed (e.g., human->gar->goldfish->zebrafish for *T3* and *I*). All BLAST hits were manually inspected for proximity to the *Tbxtb* gene. Multi-species alignments of the three elements were generated with MAFFT version 1.5.0[109].

## Identifying T-box motifs
The presence of T-box motifs in the individual species was established with FIMO version 5.5.4[46] at https://meme-suite.org/meme/tools/fimo using as input sequence the human TBXT motif *TBXT_MA0009.2.meme* obtained from JASPAR 2022[110] at https://jaspar.genereg.net/.

## Statistics and Reproducibility
The authors declare that key measures of statistics and reproducibility are built into the work throughout. For the zebrafish, axolotl, mouse, and Ciona reporter assays, as well as the mouse knockout studies, sufficient embryos were analyzed to achieve statistical significance based on previous experience in transgenic reporter assays and mouse knockout studies. Experimental sample sizes were chosen by common standards in the field and in accordance with solid phenotype designation[42,44,105,107]. For the mouse reporter assays, sample sizes were selected empirically for >3000 total putative enhancers (VISTA Enhancer Browser, https://enhancer.lbl.gov/)[111].

All transgenic reporter assays, as well as the knockout experiments, were treated with identical experimental conditions across species and performed at least twice or more times in the majority of instances. All attempts at replication were successful.

No data were excluded in the zebrafish, axolotl, mouse or Ciona reporter assays, as well as the mouse knockout studies.

Data analyses of the transgenic reporter quantification was based on injections into zebrafish, axolotl, and mouse embryos/electroporation into Ciona embryos, and knockout quantification was based

on defined genotypes of mouse embryos from crosses. No other randomizations were applicable.

Data collection for transgenic and knockout analyses was unblinded as it required reporter activity and phenotype assessment as well as genotyping analysis to confirm transgenic or mutant versus wildtype.

Zebrafish and axolotl embryos were not selected by gender as sex determination happens later in development. Ciona are hermaphroditic, therefore there is only one possible sex for individuals. Mouse embryos of both sexes were used in transgenic and knockout analyses and no differences in gender were observed in those experiments.

### Reporting summary

Further information on research design is available in the Nature Portfolio Reporting Summary linked to this article.

## Data availability

The authors declare that all the data supporting the findings of this study are available within the paper and its supplementary information files. The genome tracks using published data are deposited in a publicly accessible repository (UCSC browser). The *hg38* UCSC browser session can be found here: https://genome.ucsc.edu/cgi-bin/hgTracks?db=hg38&lastVirtModeType=default&lastVirtModeExtraState=&virtModeType=default&virtMode=0&nonVirtPosition=&position=chr6%3A166055376%2D166285375&. The *hg38* UCSC browser session can be found here: https://genome.ucsc.edu/cgi-bin/hgTracks?db=hg38&lastVirtModeType=default&lastVirtModeExtraState=&virtModeType=default&virtMode=0&nonVirtPosition=&position=chr6%3A166055376%2D166285375&hgsid=1668196600_TyrXKpANjNuIeK9hJyKBqwmyA2yAhgsid=1668196600_TyrXKpANjNuIeK9hJyKBqwmyA2yA. The *hg19* UCSC browser session can be found here: https://genome.ucsc.edu/cgi-bin/hgTracks?db=hg19&lastVirtModeType=default&lastVirtModeExtraState=&virtModeType=default&virtMode=0&nonVirtPosition=&position=chr6%3A166464129%2D166694128&hgsid=1668176188_UwkZBA1qkTeo3E3sOlYoMYl3FJC3. The mouse (*mm10*) UCSC browser session can be found here: https://genome.ucsc.edu/cgi-bin/hgTracks?db=mm10&lastVirtModeType=default&lastVirtModeExtraState=&virtModeType=default&virtMode=0&nonVirtPosition=&position=chr17%3A8368806%2D8468805&hgsid=1670749280_ioGL9AfZ5ZfCwVzWxcAwM4s0PHxk. The Monodelphis (*monDom5*) UCSC browser session can be found here: https://genome.ucsc.edu/cgi-bin/hgTracks?db=monDom5&lastVirtModeType=default&lastVirtModeExtraState=&virtModeType=default&virtMode=0&nonVirtPosition=&position=chr2%3A449921917%2D450073916&hgsid=1668178122_QQzeb4abeiOPvFBIo1AeXQ56AAQr. The spotted gar (*GCF_000242695.1*) UCSC browser session can be found here: https://genome.ucsc.edu/cgi-bin/hgTracks?db=hub_2243239_GCF_000242695.1&lastVirtModeType=default&lastVirtModeExtraState=&virtModeType=default&virtMode=0&nonVirtPosition=&position=chrLG16%3A15070915%2D15148914&hgsid=1668181420_WCqDJoX4D5OWvt0W5P7oYAFrAjcN. The zebrafish (*danRer11*) UCSC browser session can be found here: https://genome.ucsc.edu/cgi-bin/hgTracks?db=danRer11&lastVirtModeType=default&lastVirtModeExtraState=&virtModeType=default&virtMode=0&nonVirtPosition=&position=chr13%3A4394240%2D4472239&hgsid=1668178552_e2IT5zOlZFk3BhQoKpd0yek6naG5. Plasmids, stable transgenic zebrafish lines, and mouse knockout lines are available from the corresponding authors upon reasonable request.

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

## Acknowledgements

We thank Christine Archer, Molly Waters, Nikki Tsuji, Ainsley Gilbard, and Olivia Gomez (CU Anschutz), as well as Vesna Barros, Lukas Obernosterer, and Yorgos Panayotu (UZH) for excellent zebrafish husbandry support; Jitka Lachova (IMG) for outstanding expert assistance with mouse embryo preparation and strain maintenance; Beate Gruhl and Anja Wagner (TU Dresden) for excellent axolotl husbandry support; Dr. Alexa Sadier and Dr. Karen Sears (UCLA) for generously providing us with Monodelphis genomic DNA; Dr. Jelena Kresoja and Elena Cabello for bioinformatics input; and all members of our individual laboratories for critical input and discussion on experiments, concepts, and the manuscript. The species silhouettes were adapted from the PhyloPic database (www.phylopic.org). This work has been supported by an UZH URPP "Translational Cancer Research" seed grant and NIH/NIDDK grant 1R01DK129350-01A1 to A.B.; the Children's Hospital Colorado Foundation, National Science Foundation Grant 2203311, and Swiss National Science Foundation (SNSF) Sinergia grant CRSII5_180345 to C.M.; a project grant from the Swiss Cancer League, the SwissBridge Award 2016 from the SwissBridge Foundation, Additional Ventures SVRF2021-1048003 grant, and the University of Colorado Anschutz Medical Campus to C.M. and A.B.; NIH/NIGMS 1T32GM141742-01, 3T32GM121742-02S1, and NIH/NHLBI F31HL167580 to H.R.M.; Czech Science Foundation grant 23-07056 S to Z.K. and Czech Center for Phenogenomics infrastructure support LM2018126, OP VaVpI CZ.1.05/2.1.00/19.0395, and CZ.1.05/1.1.00 /02.0109 to IMG; NIH grant 5R01DE024745-09 to L.S.; NSF EDGE Award #2029216 to I.B.; SNSF Eccellenza professorship PCEFP3_186993 to M.O.; an Alexander von Humboldt postdoctoral Fellowship to A.C.; Deutsche Forschungsgemeinschaft (DFG) Grants 22137416, 450807335 and 497658823, as well as TUD and CRTD funds to M.Y.; NIH grant DP2HG010013 to E.K.F and F.L.; NIH grants R01HG003988, R01DE028599, and R01HL162304 to A.V. and B.J.M.

## Author contributions

C.L.K., H.R.M., S.B., C.M. and A.B. zebrafish experiment design and performance, UCSC browser data analysis, manuscript writing. D.K., A.C. and M.Y. axolotl experiment design and performance, data analysis, manuscript writing. B.J.M., V.R., A.V. and M.O. mouse enhancer testing experiment design and performance, data analysis, manuscript writing. V.H.A. and L.S. design and performance of experiments to assess enhancer activity on mouse embryonic sections, histology, data analysis, manuscript writing. F.L. and E.F. Ciona experiment design and performance, data analysis, manuscript writing. J.S. and Z.K. mouse CRIPSR-Cas9 knockout experiment design and performance, data analysis, manuscript writing. O.E.F. and I.B. bridge species research design, data analysis, manuscript writing.

## Competing interests

The authors declare no competing interests.

## Additional information

[1]Section of Developmental Biology, Department of Pediatrics, University of Colorado Anschutz Medical Campus, Aurora, CO, USA. [2]Institute of Molecular Genetics of the ASCR, v. v. i., Prague, Czech Republic. [3]Environmental Genomics and Systems Biology Division, Lawrence Berkeley National Laboratory, Berkeley, CA, USA. [4]Comparative Biochemistry Program, University of California, Berkeley, CA 94720, USA. [5]Technische Universität Dresden, CRTD Center for Regenerative Therapies Dresden, Dresden, Germany. [6]Department of Medicine, Health Sciences, University of California San Diego, La Jolla, CA, USA. [7]Department of Molecular Biology, Biological Sciences, University of California San Diego, La Jolla, CA, USA. [8]Biological Sciences Graduate Program, University of California San Diego, La Jolla, CA, USA. [9]Program in Craniofacial Biology, University of California San Francisco, San Francisco, CA, USA. [10]Institute for Human Genetics, University of California San Francisco, San Francisco, CA, USA. [11]Department of Orofacial Sciences, University of California San Francisco, San Francisco, CA, USA. [12]Department of Anatomy, University of California San Francisco, San Francisco, CA, USA. [13]Department for BioMedical Research (DBMR), University of Bern, Bern, Switzerland. [14]Department of Integrative Biology and Ecology, Evolution and Behavior Program, Michigan State University, East Lansing, MI, USA. [15]Institute of Molecular Life Sciences, University of Zurich, Zurich, Switzerland. [16]Max Planck Institute for Molecular Cell Biology and Genetics, Dresden, Germany. [17]Cluster of Excellence Physics of Life, Technische Universität Dresden, Dresden, Germany. [18]US Department of Energy Joint Genome Institute, Lawrence Berkeley National Laboratory, Berkeley, CA, USA. [19]School of Natural Sciences, University of California Merced, Merced, CA, USA. [20]Department of Cardiology, Bern University Hospital, Bern, Switzerland. [21]These authors contributed equally: Cassie L. Kemmler, Jana Smolikova, Hannah R. Moran. ✉e-mail: zbynek.kozmik@img.cas.cz; alexa.burger@cuanschutz.edu

