## [Peer Review File · Nature Communications]

Conserved enhancers control notochord expression of vertebrate BrachyuryREVIEWER COMMENTS

Reviewer #1 (Remarks to the Author):

I'm not sure that I would be as enthusiastic about the conserved enhancers in any other gene, but brachyury is such an important transcription factor for notochord, that I think this paper will be high interest for many readers.

Reviewer #2 (Remarks to the Author):

In this study, Kemmler and colleagues broaden our knowledge of the regulation of Brachyury TBXT expression, which is a transcription factor (TF) important for notochord and primitive streak formation, as well as tailbud specification. They identify three regulatory elements (T3, C and I) surrounding TBXT that are capable of driving reporter expression in the notochord of several chordates. The study starts by identifying these elements in chromatin accessible regions and/or bound by TBXT in chordoma tumor cells lines and ESC cells. Through extensive experiments with different animal models, the authors show that the activity of these elements is conserved in mammals. The authors also show that elements T3 and I are also conserved in more distant chordates like zebrafish, Axolott, and Ciona. Interestingly, the C region alone cannot drive TBXT-like expression in mouse and gar suggesting that it may not perform classical enhancer functions. Finally, the knockout of these elements impacts T expression in the notochord and causes defects in axial development. This observation is particularly striking when the three regions are deleted in comparison to only one or two regions deleted at the same time, which highlights the high degree of robustness provided by having three different elements contributing to T expression.

Overall, this work is well structured and very thorough. However, there are a few issues that, in the opinion of this reviewer, should be addressed for a better understanding of the paper:

1. The introduction could be more straightforward while at the same time better highlight the main goals of this work. In my opinion, the two paragraphs that focus on the impact of understanding the regulation of TBXT gene expression during human health distract from the main questions addressed in this study.
2. Could the authors explain the choice to not investigate the entire super-enhancer show in chordoma cells in this study? It seems conserved in mouse, it is bound by T, and therefore it could also show regulatory activity. Even if the entire SE couldn't be use in transgenic experiments it could have been trimmed into additional parts.
3. Does genetic data identify any variants in these three regulatory regions in patients suffering of spine abnormalities or chordoma tumors? This information could be relevant to understand how these regions regulate T expression.
4. In Fig 2, the authors provide information of the T binding sites in the three enhancer regions. Are there any other interesting TF binding motifs in these three regions, that may also explain the regulation of TBXT gene? Specifically, what could drive T expression in the C region?
5. Could the authors explain the use of ESC cells to functionally annotate the regulatory regions of TBXT ? Normally, T is only expressed in ES cells that start mesendoderm

differentiation and therefore both the ATAC-seq and T binding may not be pointing us in the right direction. Also, is the super enhancer identified in the chordoma tumor cells, also present in the ESC cells ?

6. The mouse C region is not sufficient to recapitulate TBXT expression in the mouse and gar notochord. It is also interesting that this is remarkably different from the human C region. Are there any human specific motifs that may explain these different observations? In fact, at least in the mouse it doesn't seem clear that the C region is required for any purpose. It would be important to quantitatively compare expression of T in $\Delta C/\Delta T3, C, I$ embryos against $\Delta C, I/\Delta T3, C, I$ embryos to understand whether the C region indeed contributes at all to T expression.

7 Ultimately, I cant say I understand what the authors mean by "Conserved enhancer logic" in their title. With this study, we understand much better the conservation of the activity of regulatory elements controlling T across several chordates but isn't clear to me what the term logic refers to in the context of their experiments. In fact, it is hardly used throughout the study. Perhaps a different title would be better by focusing on the surprising conservation of apparently redundant regulatory elements across chordates.

Other minor comments:

Visualizing the T ChIP-seq signal rather than the peak location would be important.

Page 4: A comment needs to be removed (Supplemental Fig. 2 new; need to edit following Supplemental Figs.)

Page 5: Could authors verify this sentence ? Wouldnt peri or post-implantation make more sense? « In mice, homozygous Brachyury/T/Tbxtb mutations in the gene body cause preimplantation defects leading to embryonic lethality between E9.5 and E10.5. »

The functional annotation of the loci in human (Fig1.A) and mouse (Fig3.A) genome could benefit from the inclusion of other marks associated with enhancer marks such as H3K27ac.

Could the authors add the conservation of the three human regions in all species studied in the study in figure 1, specifically Zebrafish and Axolotl.

Reviewer #3 (Remarks to the Author):

In this manuscript Kemmler et al explore the regulatory elements that direct the expression of the ancient T-box gene Brachyury to the notochord - a key structure in patterning chordate tissues during development. In adult vertebrates the notochord forms the nucleus pulposus of the intervertebral discs and through a process that is not well characterised, abnormal notochord derivatives may seed chordomas.

The key findings reported are that 1. Three enhancer elements together, and partially redundantly, drive Brachyury expression in the notochord, via autoregulation. One of these, TNE/T3 was previously characterised in mouse and drives notochord expression but is not required for normal development. A second is contained within an amplified region in chordomas. 2. The three elements characterised here are conserved in jawed vertebrates.

This conclusion required some evolutionary sleuthing since one of the zebrafish elements is missing, but present in another slow-evolving fish species, the gar. 3. Specific deletion of all three elements in mice results in loss of Brachyury expression in the notochord but not primitive streak and tail bud, and thus separates the phenotypic consequences of Brachyury loss in the notochord from those in the primitive streak and tail bud.

This is a nicely executed study that examines the locus in a wide range of species to draw its evolutionary conclusions, and these, together with the findings on the redundancy, autoregulation and requirement for these enhancers in mouse are well supported by data.

Major points

The novelty of the findings is slightly dimmed by two factors: 1. the enhancer that seems to be the most important for Brachyury expression in the notochord in mouse, TNE/T3, has previously been reported. It's strange that this homozygous deletion on its own is not included anywhere in this manuscript amongst the otherwise-comprehensive deletion series shown in figure 5 and S5 – would it not be important to confirm (or otherwise) the previously reported findings? 2. Perhaps because of the way in which the enhancers were identified, via a combination of sequence conservation and Brachyury binding, the only elements identified have Tbox motifs. The actual elements of enhancer logic identified are limited to positive autoregulation by Brachyury – good as far as it goes, but again this has previously been postulated (although it must be said that this study goes much further than previous ones). Furthermore, the deep sequence conservation beyond the Tbox motifs in all three enhancers points to regulatory logic that extends beyond Brachyury binding, and indeed the deletion of this motif in the C-short enhancer element doesn't abolish notochord expression in zebrafish, so there must be further elements of this regulatory logic.

It's also a little disappointing that some of the species-specific differences between enhancer regulation are unexplained, where a major emphasis of this manuscript is uncovering enhancer logic. For example, it looks like T3/TNE drives particularly high expression in posterior notochord in mouse and axolotl, but not zebrafish. Why? Why does opossum enhancer C uniquely confer expression in the Ciona notochord? Are there any other recognisable motifs present in the conserved regulatory sequence?

Perhaps in the interests of not overclaiming or overcomplicating, the manuscript is mostly silent about the initiation of reporter expression (there are a few examples in the supplementary data but these are not very well annotated: where is dorsal/ventral? Does expression correspond to shield etc? There are no images of early expression in mouse, for example. It would be informative to know if for example TNE/T3 confers earlier expression earlier than the other enhancers, perhaps initiating rather than enforcing expression? That might provide a nice explanation for its apparently greater importance for notochord expression of T, despite its more limited domain of expression regulation.

Finally, organisationally, the manuscript looks a little repetitive and perhaps not completely intuitive – although there is new information in each figure, is it necessary to have three full main figures showing human, mouse and marsupial regulatory elements driving notochord expression in other species (given that the manuscript is not primarily trying to document the small nuances of species differences)? Figure 6 looks a little out of sequence to me – it seems to belong with the evolutionary control of enhancer regulation, ie before Figure 5.

Minor points

It would be useful to include in Fig 1 that T3 is the equivalent of TNE.

Please clarify the criteria for including putative enhancer sequences. Were K and L included because they are conserved in mouse OR marsupial, or is there conservation in both that is not obvious from the sequence traces?

P4 looks like there is a phrase from the draft: 'need to edit following supp figures'

P6 I think 'preimplantation' should read 'postimplantation'

Kemmler et al. Responses to reviewer comments:

Foremost, we want to thank all reviewers for their valuable comments, and we hope you enjoy reading a revised version. We have edited the text and figures accordingly, with revisions marked in yellow in the revised manuscript.

Reviewer #1 (Remarks to the Author):

I'm not sure that I would be as enthusiastic about the conserved enhancers in any other gene, but brachyury is such an important transcription factor for notochord, that I think this paper will be high interest for many readers.

We thank the reviewer for their positive and enthusiastic take on our work. We share the reviewer's take that our manuscript will be of wide interest to researchers across the field and will act as key reference for how a key factor in notochord and axis development is regulated.

Reviewer #2 (Remarks to the Author):

In this study, Kemmler and colleagues broaden our knowledge of the regulation of Brachyury TBXT expression, which is a transcription factor (TF) important for notochord and primitive streak formation, as well as tailbud specification. They identify three regulatory elements (T3, C and I) surrounding TBXT that are capable of driving reporter expression in the notochord of several chordates. The study starts by identifying these elements in chromatin accessible regions and/or bound by TBXT in chordoma tumor cells lines and ESC cells. Through extensive experiments with different animal models, the authors show that the activity of these elements is conserved in mammals. The authors also show that elements T3 and I are also conserved in more distant chordates like zebrafish, Axolotl, and Ciona. Interestingly, the C region alone cannot drive TBXT-like expression in mouse and gar suggesting that it may not perform classical enhancer functions. Finally, the knockout of these elements impacts T expression in the notochord and causes defects in axial development. This observation is particularly striking when the three regions are deleted in comparison to only one or two regions deleted at the same time, which highlights the high degree of robustness provided by having three different elements contributing to T expression.

Overall, this work is well structured and very thorough. However, there are a few issues that, in the opinion of this reviewer, should be addressed for a better understanding of the paper:

We highly appreciate the reviewer's enthusiastic take on our collaborative work. We have now revised our manuscript based on the reviewers' input, leading to a manuscript that we believe better conveys the overall logic of our findings and the fascinating biology underlying the *Brachyury/Tbxtb* locus.

1. The introduction could be more straightforward while at the same time better highlight the main goals of this work. In my opinion, the two paragraphs that focus on the impact of understanding the regulation of TBXT gene expression during human health distract from the main questions addressed in this study.

While we acknowledge the reviewer's opinion, we feel the information in these paragraphs is important because i) it puts our discovery into a clinical context, and ii) it gives additional context of how we selected the non-coding regions for our enhancer reporter assays. However, we have shortened the paragraphs in the introduction, as some information was indeed redundant with parts of the discussion. We hope that our revised text now is more streamlined and less distracting.

2. Could the authors explain the choice to not investigate the entire super-enhancer show in chordoma cells in this study? It seems conserved in mouse, it is bound by T, and therefore it could also show regulatory activity. Even if the entire SE couldn't be use in transgenic experiments it could have been trimmed into additional parts.

The discovery of the chordoma super-enhancer by the Schreiber lab was reported around the time we had strong data for enhancer *I*, which is within this region. A better characterization of the super-enhancer's dynamics, activity, etc. is an exciting research direction we're hoping to move into as extension to our previous chordoma work and our manuscript submitted here. Ultimately, for our manuscript, we decided not to pursue the entire super-enhancer for several reasons:

i) Our uncovered, deeply conserved enhancer *I* is within the chordoma super-enhancer and sufficient to drive notochord reporter activity in different assays.

ii) While the 3'/distal part of the human super-enhancer seems to be conserved in mouse as the reviewer noted, we did not find it to be conserved in *Monodelphis* or other species. We reasoned that if that region would be a necessary and truly fundamental mammalian enhancer, it should be conserved down to marsupials.

iii) The 3' region with mouse conservation is however different from enhancer element *L*, which is conserved in *Monodelphis* but not in mouse. Enhancer candidate *L* further contains an identified ENCODE Candidate Cis-Regulatory Element, *EH38E2523939*. We therefore tested *L* in our assays and found no activity, arguing that *I* represents the only deeply conserved notochord element in the super-enhancer region. This is further corroborated by our discovery that element *I* dates back to the last common ancestor with bony fishes.

iv) The entire super-enhancer is approximately 9.8 kb and contains several repeat regions, rendering the generation of a complete reporter challenging despite our streamlined cloning protocols. We did face already

challenges with PCR-ing several regions in this super-enhancer stretch, e.g. we originally planned to include the T ChIP-seq peak in enhancer element *L*, but had to redesign the primer as we couldn't PCR up the entire ENCODE region plus the T ChIP-seq peak.

In sum, while the chordoma super-enhancer requires further investigation, our work aimed at discovering the fundamental underlying enhancer logic of the mammalian *Brachyury* locus, which then led to the discovery of its ancient enhancer architecture. Our data argues that within the chordoma super-enhancer, *I* is the core *Brachyury*-recruiting element involved in notochord activity. The remaining sequence including enhancer candidate *L* might be auxiliary, such as involved in recruiting additional co-factors. These effects warrant targeted research efforts that are however outside our focus for our manuscript presented here.

3. Does genetic data identify any variants in these three regulatory regions in patients suffering of spine abnormalities or chordoma tumors? This information could be relevant to understand how these regions regulate *T* expression.

We are delighted that our manuscript triggered this comment from the reviewer! The reviewer raises a critical point that is the focus of our future work. Using Clinvar and Gabriella Miller Kids First databases, we sought to identify variants in our enhancers; however, these databases most exclusively contain coding region variants as the predominant sequencing data is from exomes. We are currently undergoing efforts to collaborate with colleagues at several institutions including our own Children's Hospital Colorado, St. Jude's and UCSD to investigate this further.

4. In Fig 2, the authors provide information of the T binding sites in the three enhancer regions. Are there any other interesting TF binding motifs in these three regions, that may also explain the regulation of *TBXT* gene? Specifically, what could drive *T* expression in the C region?

The reviewer raises another important point, and we have since sought to address. In *Ciona*, *Brachyury* has been shown to act together with *FoxA* and *Mnx* factors in controlling notochord-specific target genes (Reeves et al., 2021; José-Edwards et al., 2015; Passamaneck et al., 2009). Using FIMO, we have identified additional *FoxA2* and *Mnx1* sites in our enhancers, and we are currently pursuing this in a follow-up study to decode *Brachyury* target genes in later spine development (Kemmler, Jacobson, et al., ongoing). Concerning enhancer element *hs_C*, the T-box we initially deleted, T-box 184-199, with a p-value of $p < 0.005$, did not result in loss of reporter activity (which we now moved into Supplemental Fig. 2C). We have since deleted another T-box right next to the first one, T-box 201-216 with a slightly higher p-value ($p < 0.008$) (Supplemental Fig. 2D), but still observed reporter activity similar to T-box 184-199. However, when we deleted T-box 184-199 and T-box 201-216, we observed loss of reporter activity comparable to the T-box deletions in *hs_T3* and *hs_I* (now in revised Fig. 2G). There are two additional T-boxes, T-box 153-168 and T-box 218-233 with even higher, likely insignificant p-values (revised Supplemental Fig. 2) that we did not further evaluate. We have added the new data to the results part of Fig. 2 and Supplemental Fig. 2.

5. Could the authors explain the use of ESC cells to functionally annotate the regulatory regions of *TBXT*? Normally, *T* is only expressed in ES cells that start mesendoderm differentiation and therefore both the ATAC-seq and T binding may not be pointing us in the right direction.

The data of the ESCs were the only publicly available data to us that contained T ChIP-seq data. We reasoned they would be a potent source to identify putative enhancer elements, as mammalian *Brachyury* has been postulated to control its own notochord expression. Further, while ES cell differentiation does not routinely lead to any notochord fates, we also reasoned that at the initiation of mesodermal cell types, *Brachyury* will bind both streak and notochord-relevant enhancers before locking into more prominent binding depending on the differentiating fate *in vitro* – thus providing a snapshot or ChIP-seq echo of numerous early *Brachyury* binding events.

Also, is the super enhancer identified in the chordoma tumor cells, also present in the ESC cells?

We did not find any ATAC- nor T ChIP-seq peaks in that region within the ESC data, arguing that this super-enhancer is a tissue-, timing-, or context-dependent entity.

6. The mouse C region is not sufficient to recapitulate TBXT expression in the mouse and gar notochord. It is also interesting that this is remarkably different from the human C region. Are there any human specific motifs that may explain these different observations? In fact, at least in the mouse it doesn't seem clear that the C region is required for any purpose. It would be important to quantitatively compare expression of T in $\Delta C/\Delta T3, C, I$ embryos against $\Delta C, I/\Delta T3, C, I$ embryos to understand whether the C region indeed contributes at all to T expression.

Enhancer element C indeed poses a conundrum, and our manuscript provides several hints as to what this element could contribute. Importantly, enhancer element C remains conserved over tens of millions of years across individual species, so it clearly is of value to the *Brachyury/Tbxtb* locus in select animals. However, our manuscript was geared towards enhancer discovery of individual elements, which we achieved with the collection of our assays – however, a limitation of our and other similar enhancer discovery work is the detection of possible enhancer synergy when using reporter assays. Our tested enhancer element C versions spanned several hundred base pairs (e.g. human *Cshort*), and human C clearly conveyed notochord activity in our reporter assays. The C elements from other species are the same length or longer, so we don't think we cut off any obvious adjacent sequences. One hypothesis we also mention in the discussion is that enhancer element C could act as an auxiliary element involved in stabilizing promoter contacts formed by enhancer elements T3 and I, which might be necessary depending on intervening sequences such as repeats, etc. that are species-dependent. As we continue to mine the regulatory landscape of the *Brachyury* locus in the notochord and beyond, the synergy and interactions of enhancer element C with other enhancers in the region are part of our planned work.

Concerning quantitative tests, we are currently not able to pursue such experiments in mouse mutants, most critically due to financial reasons. We were unable to maintain the ΔC line alone after analyzing dozens of crosses and animals without any phenotypic consequence. Further, the individual enhancer element I knockout animals have never shown a phenotype either that we detect in the combined double enhancer $\Delta C, I$ knockout animals. Of note, our knockouts for each enhancer element only span 1.6, 1.1, and 0.5 kb ($\Delta T3$, ΔI , and ΔC , respectively), which is comparatively short for enhancer or regulatory element knockouts published in the past. We chose small knockout regions to render it unlikely that any individual knockout causes severe topological issues affecting the *Brachyury* locus overall.

In sum, the combination of our reporter tests, comparative genomics, and mouse knockout combinations reveals the existence of three enhancers of which T3 and I have deeply conserved notochord activity, while C appears to represent a diverged or auxiliary element of nonetheless ancient origin.

7. Ultimately, I can't say I understand what the authors mean by "Conserved enhancer logic" in their title. With this study, we understand much better the conservation of the activity of regulatory elements controlling T across several chordates but isn't clear to me what the term logic refers to in the context of their experiments. In fact, it is hardly used throughout the study. Perhaps a different title would be better by focusing on the surprising conservation of apparently redundant regulatory elements across chordates.

The reviewer raises a good point – "conserved enhancer logic" refers to several levels of our identified enhancers, including:

i) autoregulatory logic: the T-box motifs in the enhancer elements and overall auto-regulatory function of our identified enhancer elements;

ii) structural wiring of the enhancers T3, C, and I within the *Tbxtb* locus;

iii) evolutionary logic in maintaining the three enhancers for hundreds of millions of years.

We have now incorporated our rationale of enhancer logic more in the text to justify the title (which has been the project's running title since many years!).

Other minor comments:

Visualizing the T ChIP-seq signal rather than the peak location would be important.

We agree and have tried to visualize the tracks accordingly, yet faced issues with the available data files. The underlying data are only correctly deposited as .bed files

(<https://www.ncbi.nlm.nih.gov/geo/query/acc.cgi?acc=GSE60606>) for upload into UCSC browser, which only enables to pinpoint peak location. We nonetheless believe that peak location is valuable information, as

underlined with our enhancer discovery. Further, the T ChIP-seq data published by Pillay and colleagues are not deposited on GEO, and we had to remap them as we only had access to the compiled reads without mapping data early on in the project.

Page 4: A comment needs to be removed (Supplemental Fig. 2 new; need to edit following Supplemental Figs.)

Thank you for catching this - we sincerely apologize for this editing error and have removed the comment.

Page 5: Could authors verify this sentence ? Wouldnt peri or post-implantation make more sense? « In mice, homozygous Brachyury/T/Tbxtb mutations in the gene body cause preimplantation defects leading to embryonic lethality between E9.5 and E10.5. »

Yes, indeed! We apologize for the error and have now corrected this.

The functional annotation of the loci in human (Fig1.A) and mouse (Fig3.A) genome could benefit from the inclusion of other makers associated with enhancer marks such as H3K27ac.

Yes, we agree and added the following details:

i) We have incorporated the tracks for the layered H3K27ac marks and the ENCODE cCREs into a revised Fig. 1A.

ii) For Fig. 3A, we added a H3K27ac, a H3K4me, and a DNase track of neural tube from E11.5 C57Bl/6 (<https://screen.encodeproject.org>). We reasoned that neural tube tissue would likely also contain notochord tissue due to the extraction procedure.

iii) We have included links with all the UCSC browser sessions in the Materials and Methods.

We hope our revised figure provides an accessible, useful go-to reference for the locus.

Could the authors add the conservation of the three human regions in all species studied in the study in figure 1, specifically Zebrafish and Axolotl.

Motivated by the reviewer's input, we have now generated a revised Fig. 1A in which we included the conservation in zebrafish. However, UCSC browser and other alignment platforms, such as DCode, still use zebrafish *tbxta* instead of *tbxtb* as the mammalian *Tbxt* homolog. Therefore, the homologous sequences of mammalian *T3*, *C* and *I* are not found in zebrafish when using UCSC browser as the incorrect zebrafish track coordinates are used. We refer to that in the results part and have cited there now also Fig. 1A. Further, we hope that our new findings outlined in our manuscript will lead to re-assessment of the orthologous tracks across platforms for this critical developmental gene locus.

Unfortunately, Axolotl is not included as comparative species in UCSC browser (likely as the whole-genome sequence is still relatively new), and therefore we couldn't add this to Fig. 1A.

However, all the homologous sequences and corresponding coordinates of the three enhancers from all species, including Axolotl, can be found in Supplemental Table 4.

Reviewer #3 (Remarks to the Author):

In this manuscript Kemmler et al explore the regulatory elements that direct the expression of the ancient T-box gene *Brachyury* to the notochord - a key structure in patterning chordate tissues during development. In adult vertebrates the notochord forms the nucleus pulposus of the intervertebral discs and through a process that is not well characterised, abnormal notochord derivatives may seed chordomas.

The key findings reported are that 1. Three enhancer elements together, and partially redundantly, drive *Brachyury* expression in the notochord, via autoregulation. One of these, TNE/T3 was previously characterised in mouse and drives notochord expression but is not required for normal development. A second is contained within an amplified region in chordomas. 2. The three elements characterised here are conserved in jawed vertebrates. This conclusion required some evolutionary sleuthing since one of the zebrafish elements is missing, but present in another slow-evolving fish species, the gar. 3. Specific deletion of all three elements in mice results in loss of *Brachyury* expression in the notochord but not primitive streak and tail bud, and thus separates the phenotypic consequences of *Brachyury* loss in the notochord from those in the primitive streak and tail bud.

This is a nicely executed study that examines the locus in a wide range of species to draw its evolutionary conclusions, and these, together with the findings on the redundancy, autoregulation and requirement for these enhancers in mouse are well supported by data.

We thank the reviewer's positive take on our manuscript and their constructive input. We addressed the raised points to the best of our abilities and scope of the manuscript, as outlined below.

Major points

The novelty of the findings is slightly dimmed by two factors:

1. the enhancer that seems to be the most important for *Brachyury* expression in the notochord in mouse, TNE/T3, has previously been reported. It's strange that this homozygous deletion on its own is not included anywhere in this manuscript amongst the otherwise-comprehensive deletion series shown in figure 5 and S5 – would it not be important to confirm (or otherwise) the previously reported findings?

The reviewer raises a very valid point concerning complete phenotype documentation as presented in our story. Originally, we did not include the phenotypes of our $\Delta T3$ single knockout allele, as this is published data by Schifferl and colleagues and data we consider prior state-of-the-art in the field. Nonetheless, we do have hetero- and homozygous $\Delta T3$ single-knockout embryos from our crosses. Our deletion allele, which contains TNE but extends 5' and 3', and in addition contains ENCODE element *EM10E0632659*, extends the published TNE phenotype. Our $\Delta T3$ homozygous embryos display short tails, as previously reported, but in addition show in a small region at the caudal end of the trunk spina bifida, raising the possibility that additional events are needed for a stronger spina bifida phenotype, which is exactly what we observe in trans-heterozygous $\Delta T3/\Delta T3, C, I$ embryos, and strongest in homozygous triple $\Delta T3, C, I/\Delta T3, C, I$ knockout embryos. Motivated by the reviewer's comment, we have now added the $\Delta T3$ phenotypes from our allele into a revised Fig. 5 and Supplemental Fig. 5.

2. Perhaps because of the way in which the enhancers were identified, via a combination of sequence conservation and *Brachyury* binding, the only elements identified have Tbox motifs. The actual elements of enhancer logic identified are limited to positive autoregulation by *Brachyury* – good as far as it goes, but again this has previously been postulated (although it must be said that this study goes much further than previous ones).

Thank you for appreciating that our study goes beyond previous ones! Indeed, we include here functional, evolutionary, and mechanistic work to provide first integrated examples of this previously postulated autoregulation.

We included two enhancer elements without significant *Brachyury* binding: enhancer element *K*, which does not contain any ATAC-seq peak nor T ChIP-seq peaks, and enhancer element *J*, which only contains a T ChIP-

seq peak in ESCs. We included those as they represent conserved sequence to other mammalian genomes; however, they do not show any reporter activity in our assays. As we outline also in our response to reviewer 2, our work expands previous work in *Ciona* into vertebrates. Studies including Reeves et al., 2021; José-Edwards et al., 2015; and Passamaneck et al., 2009 showed that *Brachyury* acts together with *FoxA* and *Mnx*, providing a basic blueprint for notochord enhancer regulation. Using FIMO, we have identified additional *FoxA2* and *Mnx1* sites in our enhancers – the mechanistic work-up of the individual enhancers also in comparison to other notochord targets, as pursued by several of us in our collaborative group, are the focus of several ongoing studies in our laboratories.

Furthermore, the deep sequence conservation beyond the Tbox motifs in all three enhancers points to regulatory logic that extends beyond *Brachyury* binding, and indeed the deletion of this motif in the C-short enhancer element doesn't abolish notochord expression in zebrafish, so there must be further elements of this regulatory logic.

We completely agree with the reviewer, and also reviewer 2 has noted details about this (see also response above). We now added additional experiments to our manuscript.

i) Concerning enhancer element *hs_C*, the T-box we initially deleted, T-box 184-199, with a p-value of $p < 0.005$, did not result in loss of reporter activity (which we now moved into Supplemental Fig. 2C). We have since deleted another T-box right next to the first one, T-box 201-216 with a slightly higher p-value ($p < 0.008$) (Supplemental Fig. 2D), but still observed reporter activity similar to T-box 184-199.

ii) When we deleted T-box 184-199 and T-box 201-216, we observed loss of reporter activity comparable to the T-box deletions in *hs_T3* and *hs_I*. These data are now included in revised Fig. 2G.

iii) There are two additional T-boxes, T-box 153-168 and T-box 218-233 with even higher, likely insignificant p-values (new Supplemental Fig. 2) that we did not further evaluate. We have added the new data to the results part of Fig. 2 and Supplemental Fig. 2.

It's also a little disappointing that some of the species-specific differences between enhancer regulation are unexplained, where a major emphasis of this manuscript is uncovering enhancer logic. For example, it looks like T3/TNE drives particularly high expression in posterior notochord in mouse and axolotl, but not zebrafish. Why?

The goal of our work presented in our manuscript was to discover the essential notochord enhancers controlling *Brachyury* in mammals – through which we then discovered the deep evolutionary history of the enhancer elements *T3*, *C*, and *I* across the tetrapod lineage. Our work did not focus on the differences in expression activity/strength, heterogeneities in reporter patterns, or overall quality disparities between the enhancers, also as our methodology is restricted in this regard (see also below). We believe our finding of three deeply conserved enhancer elements provides a major advance in our understanding of how the long-elusive notochord regulation of *Brachyury* occurs. Of note, the formation of a tail as extension of the spine beyond the pelvic girdle is highly variable across the species we tested, and might contribute to differences in reporter (but also native!) enhancer activity in regards to expression levels, activity duration, and heterogeneity. Importantly, as our cross-species analyses are based on reporter tests that fund on transient, random transgene integrations, we believe that any quantitative comparisons between tester species are not possible. In addition, the involved pieces of DNA are hundreds of millions of years removed from the individual animal models, which merely captures activity and not all the nuances of temporal and quantitative activities inherent to the native loci.

Why does opossum enhancer C uniquely confer expression in the *Ciona* notochord? Are there any other recognisable motifs present in the conserved regulatory sequence?

This is arguably the most surprising finding among our cross-species testing. We have currently no explanation for this – the T-box in particular within the *Monodelphis* enhancer element C is not different to other mammals. However, as we have also observed in our previous work involving multi-species reporter tests to uncover regulatory element activity and logic (e.g. Prummel et al., 2019; Parker et al., 2019; Tzung et al., 2023; Kemmler et al., 2023), while individual elements might have no or little activity in another species, orthologous enhancers from additional species can reveal the ancestral activity given enough sampling of different species. While we do not believe that the *Ciona* result with the Marsupial enhancer indicates that this particular version

of enhancer element C represents an ancestral or old version, we see it as an example of how such iterative, broad enhancer testing can reveal ancestral activity that can be missed if only single species are sampled.

Perhaps in the interests of not overclaiming or overcomplicating, the manuscript is mostly silent about the initiation of reporter expression

We appreciate the reviewer's perceptive read of our manuscript – aiming to discover basic tissue-selective activity of enhancers. The onset of expression is an intriguing detail of individual regulatory elements, as it also serves the community to assess if individual lines could provide *in vivo* reporters for particular processes as reference frame. However, when tested in disparate species as reporters as in our study, we indeed tried to focus on keeping the relevant info in the spotlight to not overstate interpretations as the reviewer points out. As we commented on the reviewer's point above, our reporter assays are based on cross-species analyses where we transiently and randomly insert a transgene from one organism into another. Similarly, we believe that also any qualitative comparisons, such as reporter initiation, in the different tester species would overstate observed results.

(there are a few examples in the supplementary data but these are not very well annotated: where is dorsal/ventral?)

In case the reviewer is referring to Supplemental Fig. 1, we took the reviewer's point into consideration and added the following: i) We annotated anterior/posterior in the Supplemental Fig. 1D,E to orient the reader to the position of the embryo and ii) added to the figure legend in Suppl. Fig. 1D that it is a dorsal view.

Does expression correspond to shield etc?

Assuming the reviewer is referring to our zebrafish assays, reporter expression does not exclusively correspond to the shield, as these are transient injections where the injected DNA randomly integrates into the zebrafish genome. We observed reporter expression in scattered cells throughout the zebrafish embryo at 80% epiboly, but expression later became restricted to the zebrafish notochord. We have annotated the shield in Suppl. Figs. 3A,C and 4B with asterisks and clarified this also in the text for the readership.

There are no images of early expression in mouse, for example. It would be informative to know if for example TNE/T3 confers earlier expression earlier than the other enhancers, perhaps initiating rather than enforcing expression? That might provide a nice explanation for its apparently greater importance for notochord expression of T, despite its more limited domain of expression regulation.

We agree with the reviewer that it would be interesting to define notochord enhancer activities at an earlier timepoint in the mouse, e.g. E8.5. However, such experiments would require a significant amount of time to accomplish, as it involves the generation of transgenic founders for the three different elements and collection at a developmental stage that is rather challenging experimentally. We have chosen E9.5 as this timepoint represents the earliest stage commonly analyzed in the framework of the Vista enhancer browser (<https://enhancer.lbl.gov/>, Visel et al., 2007), making these results comparable with other enhancer validation studies. We are however also in the process of establishing enhancer-creERT2 transgenic lines to further spatiotemporally dissect these activities in future studies.

Finally, organisationally, the manuscript looks a little repetitive and perhaps not completely intuitive – although there is new information in each figure, is it necessary to have three full main figures showing human, mouse and marsupial regulatory elements driving notochord expression in other species (given that the manuscript is not primarily trying to document the small nuances of species differences)?

We agree that our goal was not to provide small nuances – but rather to provide the logical narrative as to how finding the human notochord enhancers led to finding them in mice, across mammals, and finally across vertebrates. Our manuscript's organization enables the reader to follow the narrative with every single figure and stand-alone panel, which is also greatly supportive of incorporating the work into presentations (e.g. teaching to describe concepts of gene-regulatory logic, enhancer distribution in a locus, evolutionary conservation). In our revised manuscript, we have streamlined several figures to make them even more

accessible. We hope the reviewer shares our commitment to providing also uninitiated readers with figures that help recapitulate the underlying thought process.

Figure 6 looks a little out of sequence to me – it seems to belong with the evolutionary control of enhancer regulation, ie before Figure 5.

We understand the reviewer's sentiment – in the project's timeline, we initially thought that our identified enhancer elements are mammalian-specific; only through bridging species we were able to find them in all jawed vertebrates. We therefore thought that this evolutionary aspect of the manuscript should go at the end, expanding the initial search that started with the long-standing question in the field as to where the notochord element(s) for mammalian *Brachyury* are located...to find that we share them with the last common ancestor of fishes as part of the *Tbxtb* gene!

Minor points

It would be useful to include in Fig 1 that T3 is the equivalent of TNE.

Motivated by the reviewer's comment, we incorporated *TNE* into Fig. 3A, together with *Tstreak*, another previously identified *Brachyury*-regulatory element/promoter, that drives primitive streak expression. Our mouse *T3* includes *TNE*, however extends 5' as well as 3', and further contains the entire ENDOCE element *EM10E0632659*.

Please clarify the criteria for including putative enhancer sequences. Were K and L included because they are conserved in mouse OR marsupial, or is there conservation in both that is not obvious from the sequence traces?

We appreciate the reviewer bringing up this point and outline the element choices in the following manuscript sections:

i) We explain inclusion criteria for all putative enhancer elements in the second paragraph in the results section. *K* was included as it is conserved in Monodelphis and *L* was included as it is part of the super-enhancer in chordoma (see also response to reviewer 2). *L* contains the sequence of ENCODE element *EH38E2523939*, which might be a regulatory element with an activity we cannot pick up in our reporter assays so far.

ii) We explain all genomic features of the human enhancer elements in Supplemental Table 1, and have now added two additional tracks with H3K27ac as per reviewer 2's suggestion and ENCODE Candidate Cis-Regulatory Elements.

P4 looks like there is a phrase from the draft: 'need to edit following supp figures'

Yes, thank you for catching this! We apologize for overseeing this phrase during manuscript editing and now removed this phrase.

P6 I think 'preimplantation' should read 'postimplantation'

Yes, indeed! We apologize for this miswording and have edited it.

REVIEWERS' COMMENTS

Reviewer #2 (Remarks to the Author):

The paper is now ready for publication.

Reviewer #3 (Remarks to the Author):

The authors have improved the manuscript in response to reviewers' comments, with some additions of data and analysis. It's mentioned in the rebuttal that cost considerations have influenced their ability to add new data and I can appreciate this. The data as they exist now are sound and the conclusions drawn are reasonable.

I really only have one remaining issue. Both reviewer 2 and I have commented in different ways that 'conserved enhancer logic' is a bit of a stretch when very little about the regulatory logic is learned other than the autoregulation by T of itself in the notochord (and this itself doesn't explain how it's initiated). However the authors argue in their response to reviewer 2 that enhancer 'logic' means not only the regulation of an enhancer's spatiotemporal activity but also the 'structural wiring' of the enhancers- meaning their physical locations on the DNA sequence in different organisms, and the 'evolutionary logic' of maintaining the enhancers through evolutionary time. I don't think that just mentioning 'conserved enhancer logic' several times in the text really answers this point. Perhaps changing 'logic' to 'regulation' would manage reader expectations?

Reviewer 2:

The paper is now ready for publication.

Thank you!

Reviewer 3:

The authors have improved the manuscript in response to reviewers' comments, with some additions of data and analysis. It's mentioned in the rebuttal that cost considerations have influenced their ability to add new data and I can appreciate this. The data as they exist now are sound and the conclusions drawn are reasonable.

We thank the reviewer for their positive take on our revised manuscript.

I really only have one remaining issue. Both reviewer 2 and I have commented in different ways that 'conserved enhancer logic' is a bit of a stretch when very little about the regulatory logic is learned other than the autoregulation by T of itself in the notochord (and this itself doesn't explain how it's initiated). However the authors argue in their response to reviewer 2 that enhancer 'logic' means not only the regulation of an enhancer's spatiotemporal activity but also the 'structural wiring' of the enhancers- meaning their physical locations on the DNA sequence in different organisms, and the 'evolutionary logic' of maintaining the enhancers through evolutionary time. I don't think that just mentioning 'conserved enhancer logic' several times in the text really answers this point. Perhaps changing 'logic' to 'regulation' would manage reader expectations?

We agree, and have edited this in the title, abstract, and throughout the manuscript text. The changes are highlighted in yellow.